# ADVERSARIAL ROBUSTNESS OF IN-CONTEXT LEARNING IN TRANSFORMERS FOR LINEAR REGRESSION

## ABSTRACT

Transformers have demonstrated remarkable in-context learning capabilities across various domains, including statistical learning tasks. While previous work has shown that transformers can implement common learning algorithms, the adversarial robustness of these learned algorithms remains unexplored. This work investigates the vulnerability of in-context learning in transformers to *hijacking attacks* focusing on the setting of linear regression tasks. Hijacking attacks are prompt-manipulation attacks in which the adversary's goal is to manipulate the prompt to force the transformer to generate a specific output. We first prove that single-layer linear transformers, known to implement gradient descent in-context, are non-robust and can be manipulated to output arbitrary predictions by perturbing a single example in the in-context training set. While our experiments show these attacks succeed on linear transformers, we find they do not transfer to more complex transformers with GPT-2 architectures. Nonetheless, we show that these transformers can be hijacked using gradient-based adversarial attacks. We then demonstrate that adversarial training enhances transformers' robustness against hijacking attacks, even when just applied during finetuning. Additionally, we find that in some settings, adversarial training against a weaker attack model can lead to robustness to a stronger attack model. Lastly, we investigate the transferability of hijacking attacks across transformers of varying scales and initialization seeds, as well as between transformers and ordinary least squares (OLS). We find that while attacks transfer effectively between small-scale transformers, they show poor transferability in other scenarios (small-to-large scale, large-to-large scale, and between transformers and OLS).

## 1 INTRODUCTION

Transformers exhibit sophisticated in-context learning capabilities across a variety of settings such as language (Brown et al., 2020), vision (Kirsch et al., 2022; Bar et al., 2022; Zhang et al., 2023), tabular data (Hollmann et al., 2022; Requeima et al., 2024; Ashman et al., 2024), reinforcement learning and robotics (Chen et al., 2021; Raparthy et al., 2023; Team et al., 2023; Elawady et al., 2024). The mechanisms underlying this behavior, however, remain poorly understood. While recent works have made progress by studying transformer behavior on supervised learning tasks, fundamental questions about how these models learn and implement algorithms in-context remain open (Anwar et al., 2024, Section 2.1).

In this work, we investigate the mechanisms of in-context learning through the lens of adversarial robustness to hijacking attacks – a threat model where an adversary manipulates examples in the in-context set to force the model to output specific target values (Qiang et al., 2023; Bailey et al., 2023). Beyond its direct practical relevance for deployed language models and emerging applications of in-context learning across various domains for sensitive applications like clinical decision-making (Nori et al., 2023) or robot control (Elawady et al., 2024), studying hijacking attacks provides a powerful tool for probing and understanding the algorithms that transformers learn to implement in-context.

Our investigation focuses on the setting of linear regression tasks, where we analyze two architecture classes: single-layer linear attention models and GPT2-style transformers. Through a combination of theoretical analysis and extensive experiments, our investigation produces four key results that

challenge current theories about in-context learning and give insights about the adversarial robustness of in-context learning in transformers:

1. We prove that single-layer linear transformers, which prior work showed implement gradient descent on in-context data (von Oswald et al., 2022; Ahn et al., 2023; Zhang et al., 2024), are fundamentally vulnerable to hijacking through perturbation of just a single token (Theorem 4.1). This vulnerability emerges precisely because these models implement gradient descent, highlighting how seemingly desirable algorithmic properties can lead to exploitable weaknesses.

2. While GPT2-style transformers are also vulnerable to hijacking attacks, we find that successful attacks against linear transformers fail to transfer to GPT2 architectures. Through careful analysis of attack transferability between different architectures and classical learning algorithms like ordinary least squares, we show that the out-of-distribution behavior of transformers is mechanistically distinct from both gradient descent and OLS – calling into question prior explanations about what algorithms these models implement to learn in-context (Garg et al., 2022; Akyürek et al., 2022).

3. We find that hijacking attacks transfer readily between smaller transformers but show poor transferability between larger transformers of identical architecture but different random seeds, providing the first evidence that architecturally identical transformers trained on the same task may learn distinct in-context learning algorithms.

4. Despite the fundamental nature of these vulnerabilities, we show that adversarial training can effectively improve robustness, with impressive generalization: training on perturbations of $K$ examples yields robustness against manipulation of $K' > K$ tokens. This is particularly surprising given the historical difficulty of achieving robustness against adaptive adversaries in regression tasks (Diakonikolas & Kane, 2019).

Our findings have important implications for multiple research communities. For those studying in-context learning, we provide evidence that existing explanations based purely on in-distribution behavior or expressivity arguments are incomplete. For the robust statistics community, we demonstrate that transformers can learn surprisingly robust algorithms through a simple training procedure. And for the security community, we highlight fundamental vulnerabilities in in-context learning that merit attention as these capabilities are deployed across an expanding range of applications. By revealing these new insights about the mechanisms and fragilities of in-context learning, our work takes an important step toward better understanding how transformers process and learn from examples. The non-universality and mechanistic distinctness we demonstrate suggests that fully characterizing these processes – even in the highly structured setting of linear regression – may be more challenging than previously appreciated.

## 2 RELATED WORKS

**In-Context Learning of Supervised Learning Tasks:**   Our work is most closely related to prior works that have attempted to understand in-context learning of linear functions in transformers (Garg et al., 2022; Akyürek et al., 2022; von Oswald et al., 2022; Zhang et al., 2024; Fu et al., 2023; Ahn et al., 2023; Vladymyrov et al., 2024). von Oswald et al. (2022) provided a construction of weights of linear self-attention layers (Schmidhuber, 1992; Katharopoulos et al., 2020; Schlag et al., 2021) that allow the transformer to implement gradient descent over the in-context examples. They show that when optimized, the weights of the linear self-attention layer closely match their construction, indicating that linear transformers implicitly perform mesa-optimization. This finding is corroborated by the works of Zhang et al. (2024) and Ahn et al. (2023). A number of works have argued that when GPT2 transformers are trained on linear regression, they learn to implement ordinary least squares (OLS) (Garg et al., 2022; Akyürek et al., 2022; Fu et al., 2023). More recently, Vladymyrov et al. (2024) show that linear transformers also implement other iterative algorithms on noisy linear regression tasks with possibly different levels of noise. Bai et al. (2024) show that transformers can perform in-context algorithm selection: choosing different learning algorithms to solve different in-context learning tasks. Other neural architectures such as recurrent neural networks have also been shown to implement in-context learning algorithms (Hochreiter et al., 2001) such as bandit algorithms (Wang et al., 2016) or gradient descent (Kirsch & Schmidhuber, 2021).

**Hijacking Attacks:** While a considerable amount of research has been conducted on the security aspects of LLMs, most of the prior research has focused on jailbreaking attacks. To the best of our knowledge, Qiang et al. (2023) is the only prior that considers hijacking attack on LLMs or transformers during in-context learning. They show that it is possible to hijack LLMs to generate unwanted target outputs during in-context learning by including adversarial tokens in the demos. He et al. (2024) also consider adversarial perturbations to in-context data, however, their goal is to simply reduce the in-context learning performance of the model in general, and not in a targeted way. Bailey et al. (2023) demonstrate that vision-language models can be hijacked through adversarial perturbations to the vision modality alone. Similar to our work, both Qiang et al. (2023) and Bailey et al. (2023) assume a white-box setup and use gradient-based methods for finding adversarial perturbations to hijack the models.

**Robust Supervised Learning Algorithms:** There are a number of frameworks for robustness in machine learning. The framework we focus on in this work is data contamination/poisoning, where an adversary can manipulate the data in order to force predictions. Surprisingly, designing efficient robust learning algorithms, even for the relatively simple setting of linear regression, has proved quite challenging, with significant progress only being made in the last decade (Diakonikolas & Kane, 2023). Different algorithms have been devised which work under a contamination model where only labels $y$ can be corrupted (Bhatia et al., 2015; 2017; Suggala et al., 2019) or when both features $x$ and labels $y$ can be corrupted (Klivans et al., 2018; Diakonikolas et al., 2019; Cherapanamjeri et al., 2020). Note that all the aforementioned work focus on hand-designing robust learning algorithms for each problem setting. In contrast, we are concerned with understanding the propensity of the transformers to learn to implement robust learning algorithms.

There are a number of other related frameworks for robustness in machine learning, e.g., robustness with respect to imperceptible (adversarial) perturbations of the input (Goodfellow et al., 2015; Madry et al., 2018). We do not focus on these attack models in this work.

## 3  PRELIMINARIES

In this work, we investigate whether the learning algorithms that transformers learn to implement in-context are adversarially robust. We focus on the setting of in-context learning of linear models, a setting studied significantly in recent years (Garg et al., 2022; Akyürek et al., 2022; von Oswald et al., 2022; Zhang et al., 2024; Ahn et al., 2023). We assume pre-training data that are sampled as follows. Each linear regression task is indexed by $\tau \in [B]$, with each task consisting of $N$ labeled examples $(x_{\tau,i}, y_{\tau,i})_{i=1}^N$, query example $x_{\tau,\text{query}}$, parameters $w_\tau \overset{\text{i.i.d.}}{\sim} \mathsf{N}(0, I_d)$, features $x_{\tau,i}, x_{\tau,\text{query}} \overset{\text{i.i.d.}}{\sim} \mathsf{N}(0, I_d)$ (independent of $w_\tau$), and labels $y_{\tau,i} = w_\tau^\top x_{\tau,i}$, $y_{\tau,\text{query}} = w_\tau^\top x_{\tau,\text{query}}$.

The goal is to train a transformer on this data (by a method to be described shortly) and examine if, after pre-training, when we sample a new linear regression task (by sampling a new, independent $w \sim \mathsf{N}(0, I_d)$ and features $x_i$, $i = 1, \ldots, M$), the transformer can formulate accurate predictions for new, independent query examples. Note that the number of examples $M$ in a task at test time may differ from the number of examples $N$ per task observed during training.

To feed data into the transformer, we need to decide on a tokenization mechanism, which requires some care since transformers map sequences of vectors of a fixed dimension into a sequence of vectors of the same length and dimension, while the features $x_i$ are $d$-dimensional and outputs $y_i$ are scalars. That is, from a prompt of $N$ input-output pairs $(x_i, y_i)$ and a test example $x_{\text{query}}$ for which we want to make predictions, the question is how to embed

$$P = (x_1, y_1, \ldots, x_N, y_N, x_{\text{query}}),$$

into a matrix. We will consider two variants of tokenization: concatenation (denoted Concat), which concatenates $x_i$ and $y_i$ and stacks each sample into a column of an embedding matrix, and then appends $(x_{\text{query}}, 0)^\top \in \mathbb{R}^{d+1}$ as the last column:

$$E(P) = \begin{pmatrix} x_1 & x_2 & \cdots & x_N & x_{\text{query}} \\ y_1 & y_2 & \cdots & y_N & 0 \end{pmatrix} \in \mathbb{R}^{(d+1)\times(N+1)}. \qquad \text{(Concat)} \qquad (1)$$

The notation $E(P)$ emphasizes that the embedding matrix is a function of the prompt $P$, and we shall sometimes denote this as $E$ for ease of notation. This tokenization has been used in a number

of prior works on in-context learning of function classes (von Oswald et al., 2022; Zhang et al., 2024; Wu et al., 2023). Since transformers output a sequence of tokens of the same length and dimension as their input, with the Concat tokenization the natural predicted value for $x_{M+1}$ appears in the $(d+1, M+1)$ entry of the transformer output. This allows for a last-token prediction formulation of the squared-loss objective function: if $f(E; \theta)$ is a transformer, the objective function for $B$ batches of data consisting of $N+1$ samples $(x_{\tau,i}, y_{\tau,i})_{i=1}^N, (x_{\tau,\text{query}}, y_{\tau,\text{query}})$, each batch embedded into $E_\tau$, is

$$\widehat{L}(\theta) = \frac{1}{2B} \sum_{\tau=1}^B \left([f(E_\tau; \theta)]_{d+1, N+1} - y_{\tau,\text{query}}\right)^2. \tag{2}$$

We will also consider an alternative tokenization method, Interleave, where features $x$ and $y$ are interleaved into separate tokens,

$$E(P) = \begin{pmatrix} x_1 & 0 & x_2 & \cdots & x_N & 0 & x_{\text{query}} \\ 0 & y_1 & 0 & \cdots & 0 & y_N & 0 \end{pmatrix} \in \mathbb{R}^{(d+1) \times (2N+1)}. \tag{Interleave} \tag{3}$$

By using causal masking, i.e. forcing the prediction for the $i$-th column of $E_\tau$ to depend only on columns $\leq i$, this tokenization allows for the formulation of a next-token prediction averaged across all $N$ pairs of examples,

$$\widehat{L}(\theta) = \frac{1}{2B} \sum_{\tau=1}^B \frac{1}{N} \left( \sum_{i=1}^N [f^{\text{Mask}}(E_\tau; \theta)]_{d+1, 2i+1} - y_{\tau, i+1} \right)^2, \tag{4}$$

where we treat $y_{\tau, N+1} := y_{\tau, \text{query}}$. This formulation was used in the original work by Garg et al. (2022)

We consider in-context learning in two types of transformer models: single-layer linear transformers, where we can theoretically analyze the behavior of the transformer, and standard GPT-2 style transformers, where we use experiments to probe their behavior. In all experiments, we focus on the setting where $d = 20$ and the number of examples per pre-training task is $N = 40$.

### 3.1 SINGLE-LAYER LINEAR TRANSFORMER SETUP

Linear transformers are a simplified transformer model in which the standard self-attention layers are replaced by linear self-attention layers (Katharopoulos et al., 2020; von Oswald et al., 2022; Ahn et al., 2023; Zhang et al., 2024; Vladymyrov et al., 2024). In this work, we specifically consider a single-layer linear self-attention (LSA) model,

$$f_{\text{LSA}}(E; \theta) = f_{\text{LSA}}(E; W^{PV}, W^{KQ}) := E + W^{PV} E \cdot \frac{E^\top W^{KQ} E}{N}. \tag{5}$$

This is a modified version of attention where we remove the softmax nonlinearity, merge the projection and value matrices into a single matrix $W^{PV} \in \mathbb{R}^{d+1 \times d+1}$, and merge the query and key matrices into a single matrix $W^{KQ} \in \mathbb{R}^{d+1 \times d+1}$. For the linear transformer, we will assume the Concat tokenization.

Prior work by Zhang et al. (2024) developed an explicit formula for the predictions $f_{\text{LSA}}$ when it is pre-trained on noiseless linear regression tasks (under the Concat tokenization) by gradient flow with a particular initialization scheme. This corresponds to gradient descent with an infinitesimal learning rate $\frac{d}{dt}\theta = -\nabla L(\theta)$ in the infinite task limit $B \to \infty$ of the objective (11),

$$L(\theta) = \lim_{B \to \infty} \widehat{L}(\theta) = \frac{1}{2} \mathbb{E}_{w_\tau \sim \mathsf{N}(0, I), \, x_{\tau,i}, x_{\tau,\text{query}} \overset{\text{i.i.d.}}{\sim} \mathsf{N}(0, I)} \left[ ([f(E_\tau; \theta)]_{d+1, N+1} - x_{\tau,\text{query}}^\top w)^2 \right]. \tag{6}$$

### 3.2 STANDARD TRANSFORMER SETUP

For studying the adversarial robustness of the in-context learning in standard transformers, we use the same setup as described in Garg et al. (2022). Namely, we use a standard GPT2 architecture with the Interleave tokenization. We provide details on the architecture and the training setup in Appendix C.

### 3.3 HIJACKING ATTACKS

We focus on a particular adversarial attack where the adversary's goal is to hijack the transformer. Specifically, the aim of the adversary is to force the transformer to predict a specific output $y_{\mathsf{bad}}$ for $x_{\mathsf{query}}$ when given a prompt $P = (x_1, y_1, \ldots, x_M, y_M, x_{\mathsf{query}})$. The adversary can choose one or more pairs $(x_i, y_i)$ to replace with an adversarial example $(x_{\mathsf{adv}}^{(i)}, y_{\mathsf{adv}}^{(i)})$.

We characterize hijacking attacks in this work along two axes: $(i)$ the type of data being attacked $(ii)$ number of data-points or tokens being attacked. The adversary may perturb either the $x$ feature $(x_i, y_i) \mapsto (x_{\mathsf{adv}}, y_i)$, which we call `feature-attack`, or a label $y$, $(x_i, y_i) \mapsto (x_i, y_{\mathsf{adv}})$, which we refer to as `label-attack`, or simultaneously perturb the pair $(x_i, y_i) \mapsto (x_{\mathsf{adv}}, y_{\mathsf{adv}})$, which we refer to as `joint-attack`. We will primarily focus on `feature-attack` and `label-attack` as the behavior of `joint-attack` is qualitatively quite similar to `feature-attack` (see Figures 3 and 4). Furthermore, we allow for the adversary to perturb multiple tokens in the prompt $P$. A `k-token` attack means that the adversary can perturb at most $k$ pairs $(x_i, y_i)$ in the prompt.[1]

We note that hijacking attacks are different from jailbreaks. In jailbreaking, the adversary's goal is to bypass safety filters instilled within the LLM (Willison, 2023; Kim et al., 2024). A jailbreak may be considered successful if it can elicit *any* unsafe response from the LLM. While on the other hand, the goal of a hijacking attack is to force the model to generate *specific* outputs desired by the adversary (Bailey et al., 2023), which could potentially be unsafe outputs, in which case the hijacking attack would be considered a jailbreak as well. A good analogy for jailbreaks and hijack attacks is untargeted and targeted adversarial attacks as studied in the context of image classification (Liu et al., 2016).

## 4 ROBUSTNESS OF SINGLE-LAYER LINEAR TRANSFORMERS

We first consider robustness of a linear transformer trained to solve linear regression in-context. As reviewed previously in the Section 3.1, this setup has been considered in several prior works (von Oswald et al., 2022; Zhang et al., 2024; Ahn et al., 2023), who all show that linear transformers learn to solve linear regression problems in-context by implementing a (preconditioned) step of a gradient descent. We build on this prior work to show that the solution learned by linear transformers is highly non-robust and that an adversary can hijack a linear transformer with very minimal perturbations to the in-context training set. Specifically, we show that throughout the training trajectory, an adversary can force the linear transformer to make any prediction it would like by simply adding a single $(x_{\mathsf{adv}}, y_{\mathsf{adv}})$ pair to the input sequence. We provide a constructive proof of this theorem in Appendix A.

**Theorem 4.1.** *Let $t \geq 0$ and let $f_{\mathsf{LSA}}(\cdot \, ; \theta(t))$ be the linear transformer trained by gradient flow on the population loss using the initialization of Zhang et al. (2024), and denote $\theta(\infty)$ as the infinite-time limit of gradient flow. For any time $t \in \mathbb{R}_+ \cup \{\infty\}$ and prompt $P = (x_1, y_1, \ldots, x_M, y_M, x_{\mathsf{query}})$ with $x_{\mathsf{query}} \sim \mathsf{N}(0, I)$, for any $y_{\mathsf{bad}} \in \mathbb{R}$, the following holds.*

1. *If $x_{\mathsf{adv}} \sim \mathsf{N}(0, I_d)$, there exists $y_{\mathsf{adv}} = y_{\mathsf{adv}}(t) \in \mathbb{R}$ s.t. with probability 1 over the draws of $x_{\mathsf{adv}}, x_{\mathsf{query}}$, by replacing any single example $(x_i, y_i)$, $i \leq M$, with $(x_{\mathsf{adv}}, y_{\mathsf{adv}})$, the output on the perturbed prompt $P_{\mathsf{adv}}$ satisfies $\widehat{y}_{\mathsf{query}}(E(P_{\mathsf{adv}}); \theta(t)) = y_{\mathsf{bad}}$.*

2. *If $y_{\mathsf{adv}} \neq 0$, there exists $x_{\mathsf{adv}} = x_{\mathsf{adv}}(t) \in \mathbb{R}^d$ s.t. with probability 1 over the draw of $x_{\mathsf{query}}$, by replacing any single example $(x_i, y_i)$, $i \leq M$, with $(x_{\mathsf{adv}}, y_{\mathsf{adv}})$, the output on the perturbed prompt $P_{\mathsf{adv}}$) satisfies $\widehat{y}_{\mathsf{query}}(E(P_{\mathsf{adv}}); \theta(t)) = y_{\mathsf{bad}}$.*

Theorem 4.1 demonstrates that throughout the training trajectory, by adding a single $(x_{\mathsf{adv}}, y_{\mathsf{adv}})$ token an adversary can force the transformer to make any prediction the adversary would like. Moreover, the $(x_{\mathsf{adv}}, y_{\mathsf{adv}})$ pair can be chosen so that either $x_{\mathsf{adv}}$ is in-distribution (i.e., has the same distribution as the training data and other in-context examples) or $y_{\mathsf{adv}}$ is in-distribution. We provide explicit formulas for each of these attacks in the Appendix (see (17) and (18)).

---

[1]Note that for standard transformers with the Interleave tokenization, a k-token attack corresponds to $2k$ tokens being manipulated (see (3)).

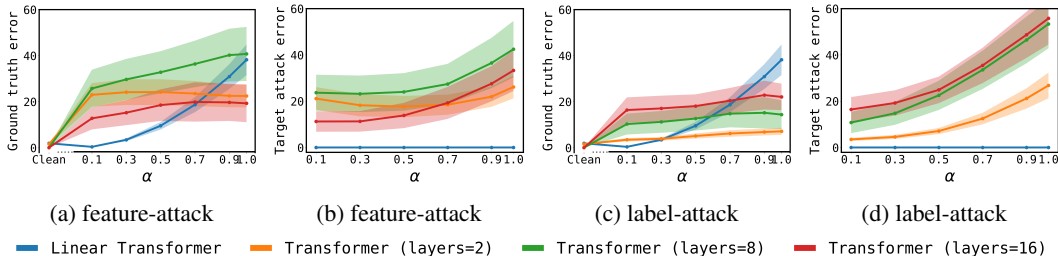

Figure 1: Robustness of different SGD-trained transformers when using attacks constructed from the gradient flow solution via Theorem 4.1, for different target values $y_{\mathsf{bad}} = (1 - \alpha)w^\top x_{\mathsf{query}} + \alpha w_\perp^\top x_{\mathsf{query}}$, where $w_\perp \perp w$. While these attacks reduce ground truth error across all model classes, the *targeted* attack error is only small for the linear transformer. Shaded area is standard error.

At a high level, the non-robustness of the linear transformer is a consequence of the linear transformer implementing a learning algorithm – one step gradient step – that generalizes well but is inherently non-robust. At a more mechanistic level, this non-robustness can be attributed to the learned in-context algorithm's inability to identify and remove outliers from the prompt. This property is shared by many learning algorithms for regression problems: for instance, ordinary least squares, as an algorithm which is linear in the labels $y$, can also be shown to suffer similar problems as the linear transformer outlined in Theorem 4.1. While non-robustness of the transformers to hijacking attacks has been established in prior works (Qiang et al., 2023; Bailey et al., 2023), this is the first result that provides a mechanistic explanation as to *why* transformers are vulnerable to hijacking attacks.

## 5 ROBUSTNESS OF STANDARD TRANSFORMERS

In this section, we empirically investigate three questions related to the robustness of GPT2-style standard transformers in this section. First, prior work has shown that when GPT2 architectures are trained on linear regression tasks, they learn to implement algorithms similar to either a single step of gradient descent (Zhang et al., 2024) or ordinary least squares (Akyürek et al., 2022; Garg et al., 2022; Fu et al., 2023). We thus examine whether the attacks from Theorem 4.1 transfer to these more complex transformer architectures. Second, we investigate gradient-based attacks on GPT2-style transformers, and whether adversarial training (during pre-training or by fine-tuning) can improve the robustness of the transformers. Third, we investigate whether gradient-based attacks transfer between different GPT2-style transformers. Unless indicated otherwise, we will be focusing the attention on a 8 layer transformer.

**Metrics**: To evaluate the impact of our adversarial attacks, we use two metrics: *ground truth error* (GTE), and *targeted attack error* (TAE). Ground-truth error measures mean-squared error (MSE) between the transformer's prediction on the corrupted prompt $P_{\mathsf{adv}}$ and the ground-truth prediction, i.e., $y_{\mathsf{clean}} = w^\top x_{\mathsf{query}}$. Targeted attack error similarly measures mean-squared error (MSE) between the transformer's prediction on the corrupted prompt and $y_{\mathsf{bad}}$. Let $\widehat{y}$ be the transformer's prediction corresponding to $x_{\mathsf{query}}$, then:

$$\text{Ground Truth Error} = \frac{1}{B} \sum_{i=1}^{B} \left( \widehat{y}_i - y_{\mathsf{clean}} \right)^2, \quad \text{Targeted Attack Error} = \frac{1}{B} \sum_{i=1}^{B} \left( \widehat{y}_i - y_{\mathsf{bad}} \right)^2. \tag{7}$$

### 5.1 DO ATTACKS FROM LINEAR TRANSFORMERS TRANSFER?

We implement separate `feature-attack` and `label-attack` based on formulas given in equations 17 and 18. Specifically, given a prompt $P = (x_1, y_1, \ldots, x_M, y_M, x_{\mathsf{query}})$, for `feature-attack`, we replace $(x_1, y_1)$ with $(x_{\mathsf{adv}}, y_1)$, and for `label-attack`, we replace $(x_1, y_1)$ with $(x_1, y_{\mathsf{adv}})$. We choose $y_{\mathsf{bad}}$ according to the following formula,

$$y_{\mathsf{bad}} = (1 - \alpha)w_\tau^\top x_{\mathsf{query}} + \alpha w_\perp^\top x_{\mathsf{query}} \tag{8}$$

Here $w_\tau$ is the underlying weight vector corresponding to the clean prompt $P$ and $w_\perp \perp w$, and $\alpha \in [0, 1]$ is a parameter. When $\alpha \to 0$, the target label $y_{\text{bad}}$ is more similar to the in-distribution ground truth, while $\alpha \to 1$ represents a label which is more out-of-distribution.

In Figure 1 we show the robustness of SGD-trained single-layer linear transformers and standard transformers of different depths as a function of $\alpha$. These results are averaged over 1000 different samples and 3 random initialization seeds for every model type (see Appendix C for further details on training). We find that the gradient flow-derived attacks transfer to the SGD-trained single-layer linear transformers, as the targeted attack error is near zero for all values of $\alpha$. Moreover, while standard (GPT2) transformers trained to solve linear regression in-context incur significant ground-truth error when the prompts are perturbed using the attacks from Theorem 4.1, these attacks are not successful as *targeted* attacks, since the targeted error is large. This behavior persists across GPT2 architectures of different depths, and suggests that when trained on linear regression tasks, GPT2 architectures do not implement one step of gradient descent, as has been suggested in some prior works (von Oswald et al., 2022; Ahn et al., 2023; Zhang et al., 2024).

## 5.2  GRADIENT-BASED ADVERSARIAL ATTACKS

In the previous subsection we found that hijacking attacks derived from the linear transformer theoretical analysis do not transfer to standard transformer architectures. In this section, we evaluate whether gradient-based optimization can be used to find appropriate adversarial perturbations for hijacking the transformer.

Specifically, we randomly select a $k_{\text{test}}$ number of input examples— where $k_{\text{test}}$ is specified beforehand—and initialize their values to zero. We then optimize these $k_{\text{test}}$ tokens by minimizing the targeted attack error, for target $y_{\text{bad}}$ from (8) for different values of $\alpha \in (0, 1]$. Both during training and testing, we set the sequence length of the transformer to be 40.

Our main results appear in Figure 3 under the label $k_{\text{train}} = 0$, which show the targeted attack error for an 8 layer transformer averaged over 1000 prompts and 3 random initialization seeds when $\alpha = 1$ from (8). We note that for `feature-attack`, an adversary can achieve a very small targeted attack error with perturbing just a single token. However, for `label-attack`, achieving low targeted attack generally requires perturbing multiple y-tokens. Note that this is in contrast with linear transformers, for which we have previously shown that hijacking is possible with perturbing just a single y-token. Finally, `joint-attack` behave in a qualitatively similar way to `feature-attack` but are slightly more effective (this is most notable for $k_{\text{test}} = 1$). Additional experiments investigating different choices of $\alpha$ appear in Appendix B.3. See Appendix C.3 for details on attack procedure.

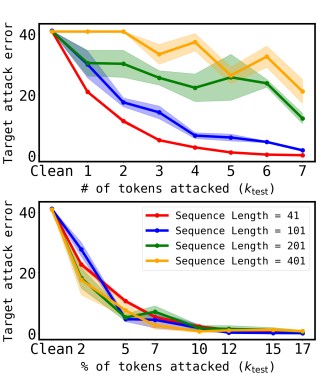

Figure 2: Larger context lengths can improve robustness for a fixed *number* of tokens attacked, but not for a fixed *proportion*. The number of layers is kept fixed at 8 while varying the context length.

## 5.3  EFFECT OF SCALING DEPTH AND SEQUENCE LENGTH

Some recent works indicate that larger neural networks are naturally more robust to adversarial attacks (Bartoldson et al., 2024; Howe et al., 2024). Unfortunately, we did not observe any consistent improvement in adversarial robustness of in-context learning in transformers in our setup with scaling of the number of layers, as can be seen in Figure 8 in the appendix.

We also studied the effect of sequence length, which scales the size of the in-context training set. We show in Figure 2 that for a fixed number of tokens attacked, longer context lengths can improve the robustness to hijacking attacks. However, for a fixed *proportion* of the context length attacked, the robustness to hijacking attacks is approximately the same across context lengths. We explore this in more detail in the appendix (see Appendix B.2).

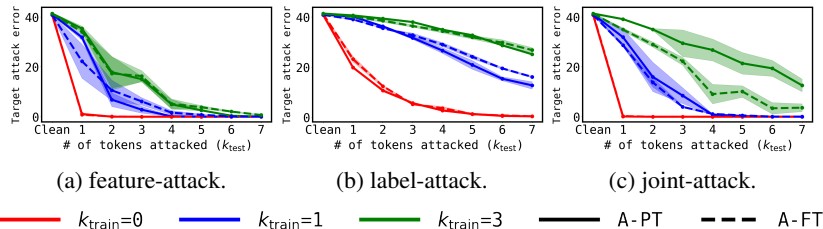

(a) feature-attack.   (b) label-attack.   (c) joint-attack.

— $k_{\text{train}}=0$  — $k_{\text{train}}=1$  — $k_{\text{train}}=3$  — A-PT  --- A-FT

Figure 3: For both adversarial pretraining (A-PT) and fine-tuning (A-FT) against `label-attack`, robustness against `label-attack` improves significantly, especially when trained on a budget of $k_{\text{train}} = 3$ perturbed tokens. The results are shown for 8 layer transformers with GPT-2 architecture.

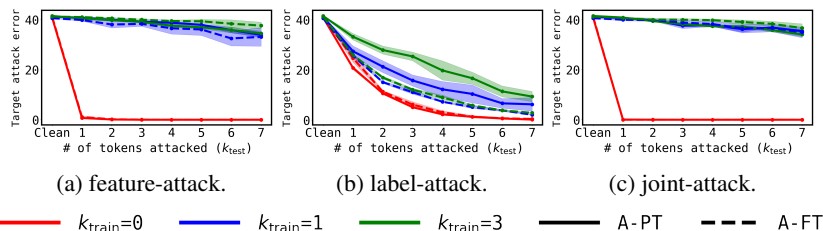

(a) feature-attack.   (b) label-attack.   (c) joint-attack.

— $k_{\text{train}}=0$  — $k_{\text{train}}=1$  — $k_{\text{train}}=3$  — A-PT  --- A-FT

Figure 4: For both adversarial pretraining (A-PT) and fine-tuning (A-FT) against `feature-attack`, robustness against `feature-attack` *and* `joint-attack` improves for 7+ token attacks when trained on $k_{\text{train}} = 1$. The results are shown for 8 layer transformers with GPT-2 architecture.

## 5.4 ADVERSARIAL TRAINING

A common tactic to promote adversarial robustness of neural networks is to subject them to adversarial training — i.e., train them on adversarially perturbed samples (Madry et al., 2018). In our setup, we create adversarially perturbed samples by carrying out the gradient-based attack outlined in Section 5.2 on the model undergoing training. Namely, for the model $f_\theta^t$ at time $t$, for each standard prompt $P$, we take a target adversarial label $y_{\text{bad}}$ and use the gradient-based attacks from Section 5.2 to construct an adversarial prompt $P_{\text{adv}}$.

We consider two types of setups for adversarial training. In the first setup, we train the transformer model from scratch on adversarially perturbed prompts. We call this *adversarial pretraining*. In the second setup, we first train the transformer model on standard (non-adversarial) prompts $P$ for $T_1$ number of steps; and then further train the transformer model for $T_2$ number of steps on adversarial prompts. We call this setup *adversarial fine-tuning*. In our experiments, unless otherwise specified, we perform adversarial pretraining for $5 \cdot 10^5$ steps. For adversarial fine-tuning, we perform $5 \cdot 10^5$ steps of standard training and then $10^5$ steps of adversarial training, i.e., $T_1 = 5 \cdot 10^5$ and $T_2 = 10^5$.

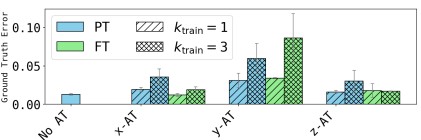

Figure 5: While there is a moderate tradeoff between robustness and (clean) accuracy when training against `label-attack`, the tradeoff is very small for `feature-attack` and `joint-attack` training.

The adversarial target value $y_{\text{bad}}$ is constructed by sampling a weight vector $w \sim \mathsf{N}(0, I)$ independent of the parameters $w_\tau$ which determine the labels for the task $\tau$ and setting $y_{\text{bad}} = w^\top x_{\text{query}}$. To keep training efficient, for each task we perform 5 gradient steps to construct the adversarial prompt. We denote the number of tokens attacked during training with $k_{\text{train}}$, and experiment with two values of $k_{\text{train}} = 1$ and $k_{\text{train}} = 3$. Unless stated otherwise, we use an 8 layer transformer.

**Adversarial training improves robustness—even with only fine-tuning.** In Figures 3 and 4, we show the robustness of transformers under $k$-token hijacking attacks when they are adversarially

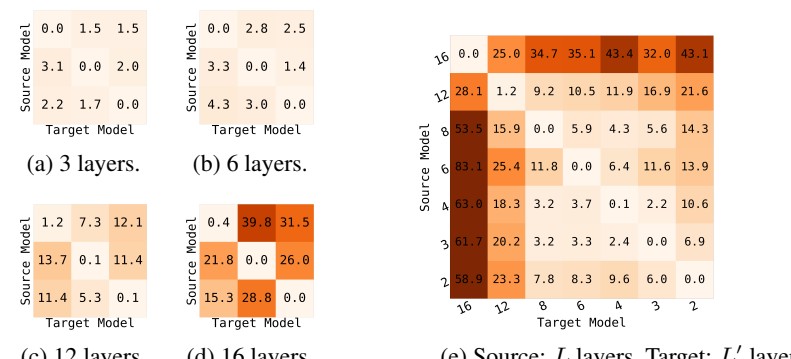

Figure 6: Targeted attack error when transferring an attack from a source model to a target models. Attacks transfer better between smaller-scale models, but not to larger-scale models (right)—even across random seeds (left). Adversarial samples were generated using `feature-attack` with $k = 3$.

trained on either `feature-attack` or `label-attack`. We see that adversarial training against attacks of a fixed type (e.g. `feature-attack` or `label-attack`) improves robustness to hijacking attacks of the same type, with robustness under feature-attack seeing a particular improvement. Notably, there is little difference between adversarial fine-tuning and pretraining, showing little benefit from the increased compute requirement of adversarial pretraining.

**Adversarial training against one attack model moderately improves robustness against another.** Following adversarial training against `label-attack`, we see modest improvement in the robustness against `feature-attack` and `joint-attack`, while adversarial training against `feature-attack` results in significant improvement against `joint-attack` (as expected, given that 20 of the 21 dimensions `joint-attack` uses is shared by `feature-attack`) and modest improvement against `label-attack`. We show in Fig. 12 the results for adversarial training against `joint-attack`.

**Adversarial training against $k$-token attacks can lead to robustness against $k' > k$ token attacks.** In both Fig. 3 and 4 (as well as Fig. 12) we see that training against $k = 3$ token attacks can lead to significant robustness against $k = 7$ token attacks, especially in the case of models trained against `feature-attack` and `joint-attack`.

**Minimal accuracy vs. robustness tradeoff.** In many supervised learning problems, there is an inherent tradeoff between the robustness of a model and its (non-robust) accuracy (Zhang et al., 2019). In Fig. 5 we compare the performance of models which undergo adversarial training vs. those which do not, and we find that while there is a moderate tradeoff when undergoing `label-attack` training, there is little tradeoff when undergoing `feature-attack` and `joint-attack` training.

On the whole, given the challenging nature of robust regression problem (Diakonikolas & Kane, 2019), the success of adversarial training is both surprising and remarkable, and hints at the ability of transformers to solve highly challenging non-convex optimization problems in context.

## 5.5 TRANSFERABILITY OF ADVERSARIAL ATTACKS ACROSS TRANSFORMERS

In this section, we evaluate how the adversarial attacks transfer between transformers. Note that we are specifically interested in *targeted* transfer; i.e., we want adversarial samples generated by attacking a source model to predict $y_{\text{bad}}$ to also cause a victim model to predict $y_{\text{bad}}$. Transfer of targeted attacks on neural networks is generally much less common than the transfer of untargeted attacks (Liu et al., 2016).

Due to space limitations we restrict our focus to `feature-attack` here; transferability of `label-attack` follows a similar pattern and is discussed in Appendix B.4. We first consider *within-class transfer*, i.e., transfer from one transformer to another transformer with identical architecture but trained from a different random initialization. In Figure 6(a-d), we see that for transformers with smaller capacities (3 and 6 layers) attacks transfer quite well, but transfers become

progressively worse as the models become larger. This suggests that higher-capacity transformers could implement different in-context learning algorithms when trained from different seeds.

We next consider *across-class transfer*, i.e. transfer between transformers with different layers. Fig. 6(e) shows a similar trend as within-class transfer: attacks from small-to-medium capacity models transfer better to other small-to-medium capacity models, while larger capacity models transfer poorly to all other capacity models.

## 5.6 TRANSFERABILITY OF ADVERSARIAL ATTACKS BETWEEN TRANSFORMERS AND LEAST SQUARES SOLVER

It has been argued that transformers trained to solve linear regression in-context implement ordinary least squares (OLS) (Garg et al., 2022; Akyürek et al., 2022). If so, adversarial (hijacking) attacks ought to transfer between transformers and OLS. In Figure 7, we show mean squared error (MSE) between predictions of OLS and transformers on adversarial samples created by performing feature-attack on OLS and transformers respectively. It can be clearly observed that as the targeted prediction $y_{\text{bad}}$ becomes more out-of-distribution ($\alpha \to 1$), MSE between predictions made by OLS and transformers also increases. Furthermore, MSE is considerably larger when adversarial samples are created by attacking transformers. This collectively indicates that the alignment between OLS and transformers is weaker out-of-distribution and that the transformers likely have additional adversarial vulnerabilities relative to OLS. We provide additional results and expanded discussion in Appendix B.5.

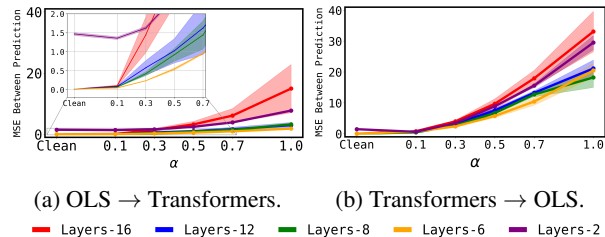

(a) OLS $\to$ Transformers.     (b) Transformers $\to$ OLS.

■ Layers-16   ■ Layers-12   ■ Layers-8   ■ Layers-6   ■ Layers-2

Figure 7: Mean squared error between predictions made by OLS and transformers on adversraial samples sourced respectively from OLS and transformers for different values of $\alpha$.

## 6 DISCUSSION & FUTURE WORK

This work has many surprising findings that provide avenues of future work. Firstly, through our analysis of transferability of adversarial attacks between GPT-2 style transformers and traditional solvers (ordinary least squares and gradient descent implemented by linear transformers), we have exposed that these transformers behave differently to these solvers out-of-distribution. This calls into question the prior explanations of in-context learning in this setting that transformers implement 'familiar algorithms' in-context (Akyürek et al., 2022; Garg et al., 2022; Zhang et al., 2024). Relatedly, we have shown that hijacking attacks do not even transfer across larger identical transformers. This is the first evidence of non-universality of in-context learning mechanisms within single architectures. Collectively, this indicates that developing a thorough understanding of in-context learning within transformers may be more challenging than previously thought, and emphasises the need of developing mechanistic understanding of these transformers.

Our work also sheds light on the mechanistic underpinnings of the adversarial non-robustness of transformers that has been demonstrated in prior works (Qiang et al., 2023; Bailey et al., 2023). Within linear transformers, we have shown that this vulnerability arises *because* linear transformers implement a standard non-robust learning algorithm. Prior works that have shown that gradient descent on neural network parameters tends to have an implicit bias towards learning solutions which generalize well but are not adversarially robust (Frei et al., 2023). Future works may investigate whether a similar bias exists regarding in-context learning algorithms discovered by transformers as well.

However, on the positive side, we have shown that adversarial training does improve robustness to hijacking attacks, and generalizes in a limited way. This is an encouraging and surprising result given that robust regression in the presence of an adaptive adversary is a highly challenging problem (Diakonikolas & Kane, 2019). Understanding and 'reverse-engineering' the algorithms that transformers implement could help provide novel insights for algorithm design.

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

APPENDIX

A   PROOFS

**Notation:**   We denote $[n] = \{1, 2, ..., n\}$. We write the inner product of two matrices $A, B \in \mathbb{R}^{m \times n}$ as $\langle A, B \rangle = \text{tr}(AB^\top)$. We use $0_n$ and $0_{m \times n}$ to denote the zero vector and zero matrix of size $n$ and $m \times n$, respectively. We denote the matrix operator norm and Frobenius norm as $\|\cdot\|_2$ and $\|\cdot\|_F$. We use $I_d$ to denote the $d$-dimensional identity matrix and sometimes we also use $I$ when the dimension is clear from the context.

**Setup:**   As described in the main text, we consider the setting of linear transformers trained on in-context examples of linear models, a setting considered in a number of prior theoretical works on transformers (von Oswald et al., 2022; Akyürek et al., 2022; Zhang et al., 2024; Ahn et al., 2023; Wu et al., 2023). Let $x_i \in \mathbb{R}^d$ and $y_i \in \mathbb{R}$. For a prompt $P = (x_1, y_1, \ldots, x_N, y_N, x_{N+1})$, we say its *length* is $N$. For this prompt, we use an embedding which stacks $(x_i, y_i)^\top \in \mathbb{R}^{d+1}$ into the first $N$ columns with $(x_{N+1}, 0)^\top \in \mathbb{R}^{d+1}$ as the last column:

$$E = E(P) = \begin{pmatrix} x_1 & x_2 & \cdots & x_N & x_{N+1} \\ y_1 & y_2 & \cdots & y_N & 0 \end{pmatrix} \in \mathbb{R}^{(d+1) \times (N+1)}. \tag{9}$$

We consider a single-layer linear self-attention (LSA) model, which is a modified version of attention where we remove the softmax nonlinearity, merge the projection and value matrices into a single matrix $W^{PV} \in \mathbb{R}^{d+1, d+1}$, and merge the query and key matrices into a single matrix $W^{KQ} \in \mathbb{R}^{d+1, d+1}$. Denote the set of parameters as $\theta = (W^{KQ}, W^{PV})$ and let

$$f_{\text{LSA}}(E; \theta) = E + W^{PV} E \cdot \frac{E^\top W^{KQ} E}{N}. \tag{10}$$

The network's prediction for the query example $x_{N+1}$ is the bottom-right entry of matrix output by $f_{\text{LSA}}$,

$$\widehat{y}_{\text{query}}(E; \theta) = [f_{\text{LSA}}(E; \theta)]_{(d+1),(N+1)}.$$

We may occasionally use an abuse of notation by writing $\widehat{y}_{\text{query}}(E; \theta)$ as $\widehat{y}_{\text{query}}(P)$ or $\widehat{y}_{\text{query}}$ with the understanding that the transformer always forms predictions by embedding the prompt into the matrix $E$ and always depends upon the parameters $\theta$.

We assume training prompts are sampled as follows.   Let $\Lambda$ be a positive definite covariance matrix.   Each training prompt, indexed by $\tau \in \mathbb{N}$, takes the form of $P_\tau = (x_{\tau,1}, h_\tau(x_{\tau_1}), \ldots, x_{\tau,N}, h_\tau(x_{\tau,N}), x_{\tau,N+1})$, where task weights $w_\tau \overset{\text{i.i.d.}}{\sim} \mathsf{N}(0, I_d)$, inputs $x_{\tau,i} \overset{\text{i.i.d.}}{\sim} \mathsf{N}(0, \Lambda)$, and labels $y_{\tau,i} = \langle w_\tau, x_i \rangle$. The empirical risk over $B$ independent prompts is defined as

$$\widehat{L}(\theta) = \frac{1}{2B} \sum_{\tau=1}^{B} \left( \widehat{y}_{\tau,N+1}(E_\tau; \theta) - \langle w_\tau, x_{\tau,N+1} \rangle \right)^2. \tag{11}$$

We consider the behavior of gradient flow-trained networks over the population loss in the infinite task limit $B \to \infty$:

$$L(\theta) = \lim_{B \to \infty} \widehat{L}(\theta) = \frac{1}{2} \mathbb{E}_{w_\tau \sim \mathsf{N}(0, I_d), \, x_{\tau,i} x_{\tau,N+1} \overset{\text{i.i.d.}}{\sim} \mathsf{N}(0, \Lambda)} \left[ (\widehat{y}_{\tau,N+1}(E_\tau; \theta) - \langle w_\tau, x_{\tau,N+1} \rangle)^2 \right] \tag{12}$$

Note that we consider the infinite task limit, but each task has a finite set of $N$ i.i.d. $(x_i, y_i)$ pairs. We consider the setting where $f_{\text{LSA}}$ is trained by gradient flow on the population loss above. Gradient flow captures the behavior of gradient descent with infinitesimal step size and has dynamics $\frac{\mathrm{d}}{\mathrm{d}t}\theta = -\nabla L(\theta)$.

We repeat Theorem 4.1 from the main section for convenience.

**Theorem 4.1.** *Let $t \geq 0$ and let $f_{\text{LSA}}(\,\cdot\,; \theta(t))$ be the linear transformer trained by gradient flow on the population loss using the initialization of Zhang et al. (2024), and denote $\theta(\infty)$ as the infinite-time limit of gradient flow. For any time $t \in \mathbb{R}_+ \cup \{\infty\}$ and prompt $P = (x_1, y_1, \ldots, x_M, y_M, x_{\text{query}})$ with $x_{\text{query}} \sim \mathsf{N}(0, I)$, for any $y_{\text{bad}} \in \mathbb{R}$, the following holds.*

1. *If $x_{\mathsf{adv}} \sim \mathsf{N}(0, I_d)$, there exists $y_{\mathsf{adv}} = y_{\mathsf{adv}}(t) \in \mathbb{R}$ s.t. with probability 1 over the draws of $x_{\mathsf{adv}}, x_{\mathsf{query}}$, by replacing any single example $(x_i, y_i)$, $i \le M$, with $(x_{\mathsf{adv}}, y_{\mathsf{adv}})$, the output on the perturbed prompt $P_{\mathsf{adv}}$ satisfies $\widehat{y}_{\mathsf{query}}(E(P_{\mathsf{adv}}); \theta(t)) = y_{\mathsf{bad}}$.*

2. *If $y_{\mathsf{adv}} \neq 0$, there exists $x_{\mathsf{adv}} = x_{\mathsf{adv}}(t) \in \mathbb{R}^d$ s.t. with probability 1 over the draw of $x_{\mathsf{query}}$, by replacing any single example $(x_i, y_i)$, $i \le M$, with $(x_{\mathsf{adv}}, y_{\mathsf{adv}})$, the output on the perturbed prompt $P_{\mathsf{adv}})$ satisfies $\widehat{y}_{\mathsf{query}}(E(P_{\mathsf{adv}}); \theta(t)) = y_{\mathsf{bad}}$.*

*Proof.* By definition, for an embedding matrix $E$ with $M + 1$ columns,

$$\widehat{y}_{\mathsf{query}}(E; \theta) = \left( (w_{21}^{PV})^\top \quad w_{22}^{PV} \right) \cdot \left( \frac{EE^\top}{M} \right) \begin{pmatrix} W_{11}^{KQ} \\ (w_{21}^{KQ})^\top \end{pmatrix} x_{\mathsf{query}}. \tag{13}$$

Due to the linear attention structure, note that the prediction is the same when replacing $(x_k, y_k)$ with $(x_{\mathsf{adv}}, y_{\mathsf{adv}})$ for any $k$, so for notational simplicity of the proof we will consider the case of replacing $(x_1, y_1)$ with $(x_{\mathsf{adv}}, y_{\mathsf{adv}})$. So, let us consider the embedding corresponding to $(x_{\mathsf{adv}}, y_{\mathsf{adv}}, x_2, y_2, \ldots, x_M, y_M, x_{\mathsf{query}})$, so that

$$EE^\top = \frac{1}{M} \begin{pmatrix} x_{\mathsf{adv}} x_{\mathsf{adv}}^\top + \sum_{i=2}^M x_i x_i^\top + x_{\mathsf{query}} x_{\mathsf{query}}^\top & y_{\mathsf{adv}} x_{\mathsf{adv}} + \sum_{i=2}^M y_i x_i \\ y_{\mathsf{adv}} x_{\mathsf{adv}}^\top + \sum_{i=2}^M y_i x_i^\top & y_{\mathsf{adv}}^2 + \sum_{i=2}^M y_i^2 \end{pmatrix}.$$

Expanding, we have

$$\widehat{y}_{\mathsf{query}}(E; \theta) = \frac{(w_{21}^{PV})^\top}{M} \left( x_{\mathsf{adv}} x_{\mathsf{adv}}^\top + \sum_{i=2}^M x_i x_i^\top + x_{\mathsf{query}} x_{\mathsf{query}}^\top \right) W_{11}^{KQ} x_{\mathsf{query}}$$

$$+ \frac{(w_{21}^{PV})^\top}{M} \left( y_{\mathsf{adv}} x_{\mathsf{adv}} + \sum_{i=2}^M y_i x_i \right) (w_{21}^{KQ})^\top x_{\mathsf{query}}$$

$$+ \frac{w_{22}^{PV}}{M} \left( y_{\mathsf{adv}} x_{\mathsf{adv}}^\top + \sum_{i=2}^M y_i x_i^\top \right) W_{11}^{KQ} x_{\mathsf{query}}$$

$$+ \frac{w_{22}^{PV}}{M} \left( y_{\mathsf{adv}}^2 + \sum_{i=2}^M y_i^2 \right) (w_{21}^{KQ})^\top x_{\mathsf{query}}.$$

When training by gradient flow over the population using the initialization of (Zhang et al., 2024, Assumption 3.3), by Lemmas C.1, C.5, and C.6 of (Zhang et al., 2024) we know that for all times $t \in \mathbb{R}_+ \cup \{\infty\}$, it holds that $w_{21}^{PV}(t) = w_{12}^{PV}(t) = w_{21}^{KQ}(t) = 0$ and $W_{11}^{KQ}(t) \neq 0$ and $w_{22}^{PV}(t) \neq 0$. In particular, the prediction formula above simplifies to

$$\widehat{y}_{\mathsf{query}}(E; \theta(t)) = \frac{w_{22}^{PV}(t)}{M} \left( y_{\mathsf{adv}} x_{\mathsf{adv}}^\top + \sum_{i=2}^M y_i x_i^\top \right) W_{11}^{KQ}(t) x_{\mathsf{query}}. \tag{14}$$

For notational simplicity let us denote $W(t) = w_{22}^{PV}(t) W_{11}^{KQ}(t)$, so that

$$\widehat{y}(E; \theta(t)) = \frac{1}{M} \left( y_{\mathsf{adv}} x_{\mathsf{adv}}^\top + \sum_{i=2}^M y_i x_i^\top \right) W(t) x_{\mathsf{query}}.$$

The goal is to take $y_{\mathsf{bad}} \in \mathbb{R}$ and find $(x_{\mathsf{adv}}, y_{\mathsf{adv}})$ such that $\widehat{y}(E; \theta(t)) = y_{\mathsf{bad}}$. Rewriting the above equation we see that this is equivalent to finding $(x_{\mathsf{adv}}, y_{\mathsf{adv}})$ such that

$$y_{\mathsf{adv}} x_{\mathsf{adv}}^\top W(t) x_{\mathsf{query}} = M \left( y_{\mathsf{bad}} - \frac{1}{M} \sum_{i=2}^M y_i x_i^\top W(t) x_{\mathsf{query}} \right). \tag{15}$$

From here we see that if $W(t) x_{\mathsf{query}} \neq 0$ then by setting

$$x_{\mathsf{adv}} y_{\mathsf{adv}} = \frac{M W(t) x_{\mathsf{query}}}{\|W(t) x_{\mathsf{query}}\|^2} \cdot \left( y_{\mathsf{bad}} - \frac{1}{M} \sum_{i=2}^M y_i x_i^\top W(t) x_{\mathsf{query}} \right), \tag{16}$$

we guarantee that $\widehat{y}(E; \theta(t)) = y_{\mathsf{bad}}$. By Zhang et al. (2024, Lemmas A.3 and A.4), we know $W(t) \neq 0$ for all $t$. Since $W(t) \neq 0$ and $x_{\mathsf{query}} \sim \mathsf{N}(0, I)$ is independent of $W(t)$, we know $W(t)x_{\mathsf{query}} \neq 0$ a.s. Therefore the identity (16) suffices for constructing adversarial tokens, and indeed for any choice of $y_{\mathsf{adv}} \neq 0$ this directly allows for constructing $x$-based adversarial tokens,

$$x_{\mathsf{adv}} = \frac{MW(t)x_{\mathsf{query}}}{y_{\mathsf{adv}}\|W(t)x_{\mathsf{query}}\|^2} \cdot \left( y_{\mathsf{bad}} - \frac{1}{M}\sum_{i=2}^{M} y_i x_i^\top W(t)x_{\mathsf{query}} \right), \tag{17}$$

On the other hand, if we want to construct an adversarial token by solely changing the label $y$, we can return to (15). Clearly, as long as $x_{\mathsf{adv}}^\top W(t)x_{\mathsf{query}} \neq 0$, then dividing both sides by this quantity allows for solving $y_{\mathsf{adv}}$. If we assume $x_{\mathsf{adv}}$ is another in-distribution independent $N(0, I)$ sample, then since $W(t) \neq 0$ guarantees that $x_{\mathsf{adv}}^\top W(t)x_{\mathsf{query}} \neq 0$ and so we can construct

$$y_{\mathsf{adv}} = \frac{M\left( y_{\mathsf{bad}} - \frac{1}{M}\sum_{i=2}^{M} y_i x_i^\top W(t)x_{\mathsf{query}} \right)}{x_{\mathsf{adv}}^\top W(t)x_{\mathsf{query}}}. \tag{18}$$

$\square$

## B ADDITIONAL RESULTS

### B.1 EFFECT OF SCALE

We conducted experiments with transformers with different number of layers to evaluate whether scale has any effect on adversarial robustness of the transformer or not. We observed no meaningful improvement in the adversarial robustness of the transformers with increase in the number of layers. This is shown in the figure below for $y_{\mathsf{bad}}$ chosen with $\alpha = 1$. See Section 5.3 in the main text for relevant discussion.

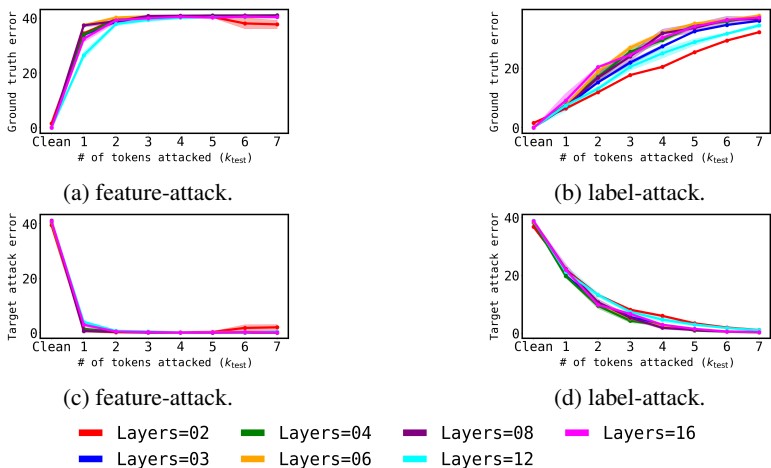

Figure 8: Increasing the scale of the transformer does not improve the adversarial robustness of in-context learning in transformers.

### B.2 EFFECT OF SEQUENCE LENGTH

We show here the complete set of results, for both `feature-attack` and `label-attack`, on how an increase in sequence length positively impacts adversarial robustness if adversary can manipulate the same number of tokens (for all sequence lengths), but if the adversary can manipulate the same proportion of tokens (which would amount to different number of tokens for different sequence lengths), increase in sequence length has a negligble effect on the adversarial robustness. See Section 5.3 in the main text for relevant discussion.

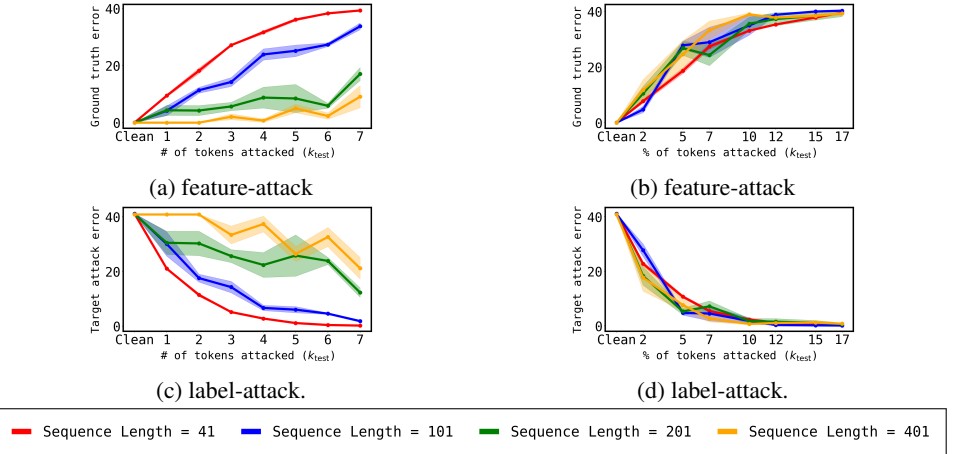

Figure 9: Effect of increase in sequence length.

### B.3 GRADIENT-BASED ADVERSARIAL ATTACKS & ADVERSARIAL TRAINING

In the main text (in Sections 5.2 and 5.4), we gave results for attacks performed with $y_{bad}$ chosen by setting $\alpha = 1$ in equation 8. Here, we present results for $\alpha = 0.5$ and $\alpha = 0.1$. These results are qualitatively similar to the case of $\alpha = 1$ and are presented only for completeness. Furthermore, in the main text, we showed only target attack error for our attacks due to space constraints, while here we present results for both ground truth error and target attack error.

#### B.3.1 $\alpha = 1.0$

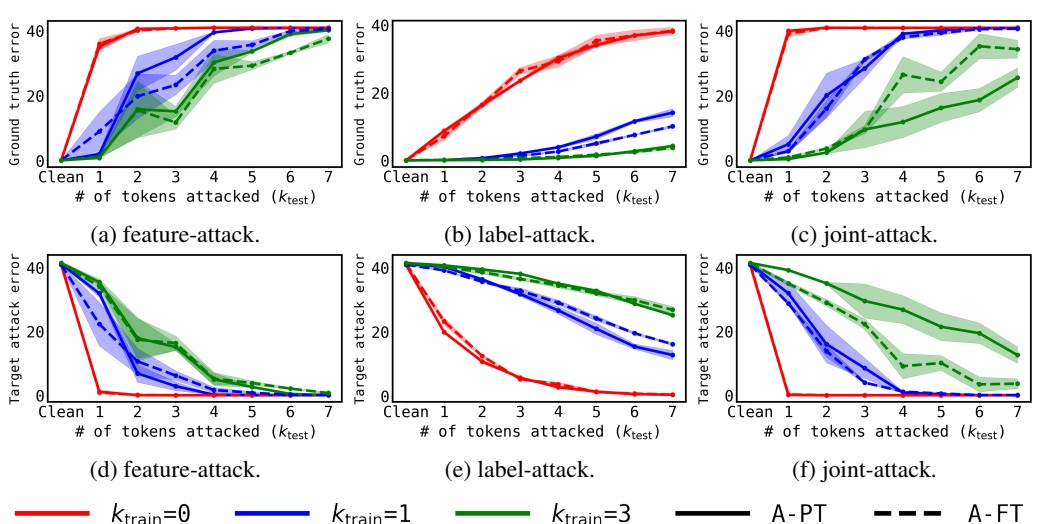

Figure 10: Adversarial training against `label-attack`. A-PT denotes adversarial pretraining and A-FT denotes adversarial finetuning. $k_{train}$ denotes the number of tokens attacked during training and $k_{train} = 0$ corresponds to a model that has not undergone adversarial training at all.

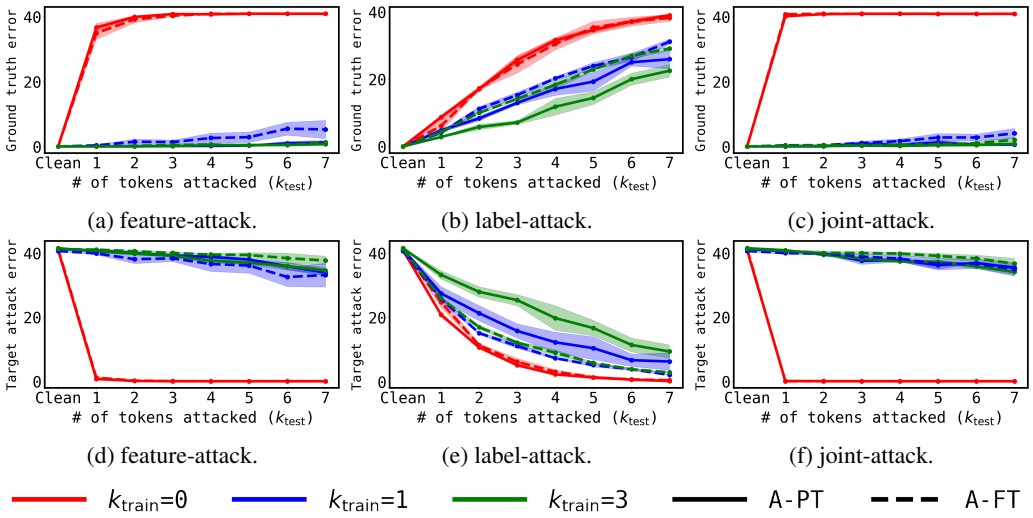

Figure 11: Adversarial training against `feature-attack`.

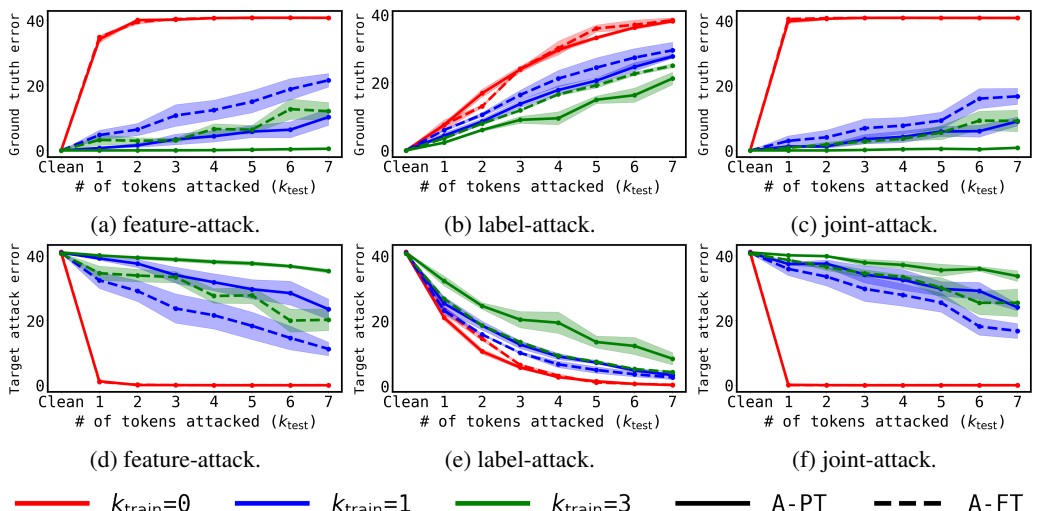

Figure 12: Adversarial training against `joint-attack`.

### B.3.2  $\alpha = 0.5$

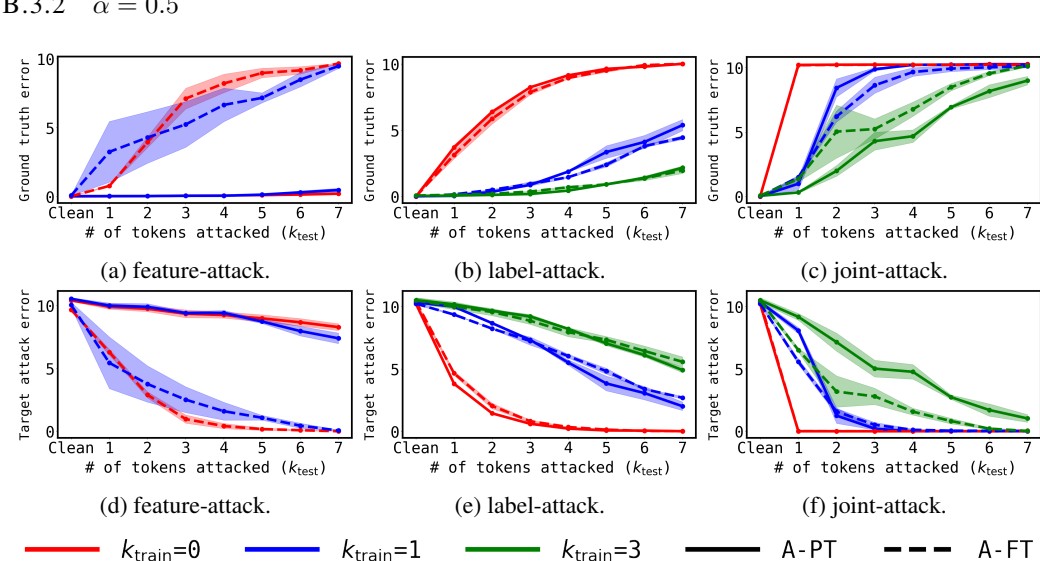

Figure 13: Adversarial training against `label-attack`.

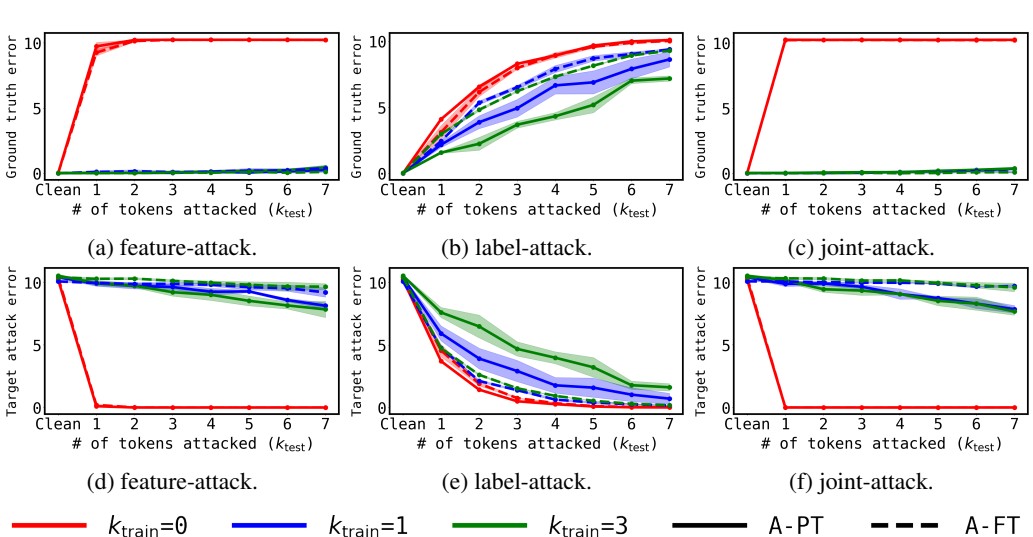

Figure 14: Adversarial training against `feature-attack`.

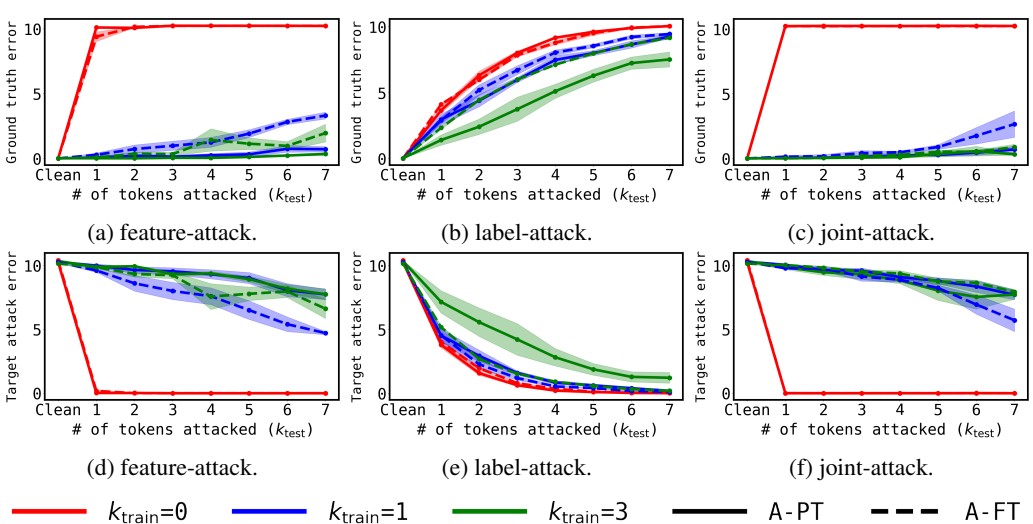

Figure 15: Adversarial training against `joint-attack`.

### B.3.3 $\alpha = 0.1$

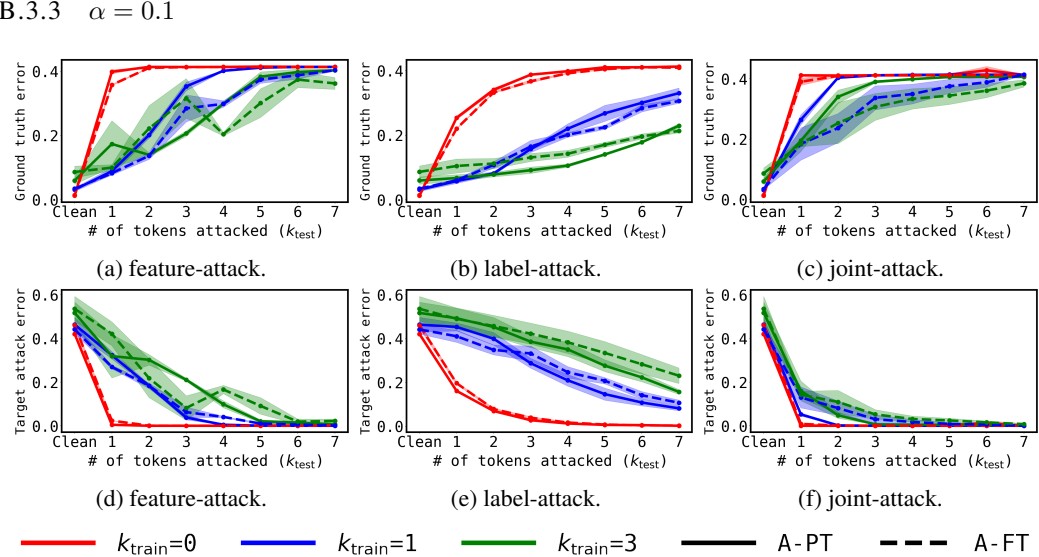

Figure 16: Adversarial training against `label-attack`. A-PT denotes adversarial pretraining and A-FT denotes adversarial finetuning. $k_{\text{train}}$ denotes the number of tokens attacked during training and $k_{\text{train}} = 0$ corresponds to a model that has not undergone adversarial training at all.

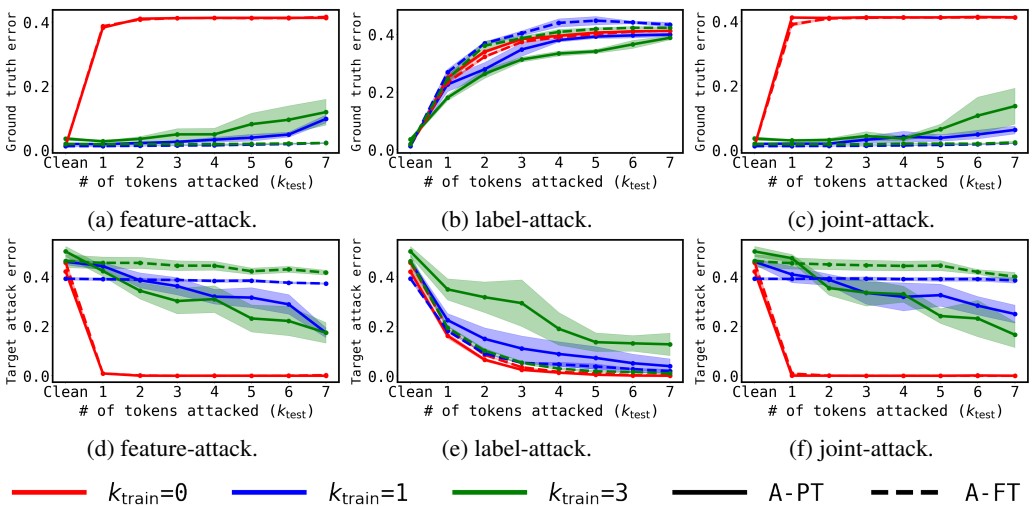

Figure 17: Adversarial training against `feature-attack`.

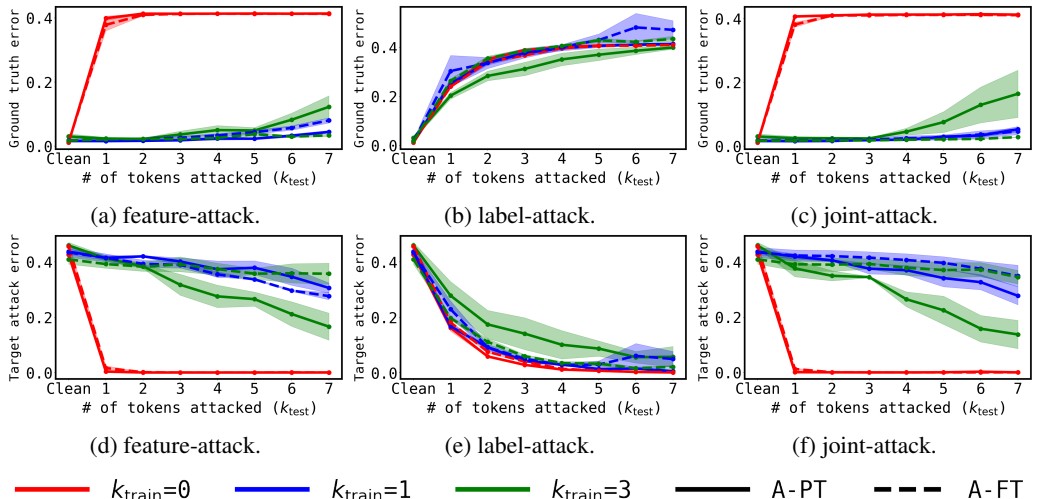

Figure 18: Adversarial training against `joint-attack`.

## B.4 TRANSFERABILITY

In Section 5.5, we briefly presented some results around transfer of adversarial examples generated using one transformer to other transformers – either with the same architecture or different architecture. We present complete results here, for both `feature-attack` and `label-attack`. As in the main text, we first present results for transfer across same class of transformers, i.e., transformers with same number of layers and then present results for transfer across different classes of transformers.

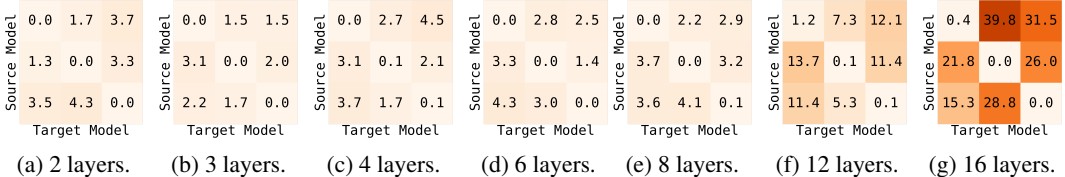

(a) 2 layers.    (b) 3 layers.    (c) 4 layers.    (d) 6 layers.    (e) 8 layers.    (f) 12 layers.    (g) 16 layers.

Figure 19: *Target Attack Error* for different target models on adversarial samples generated using a source model with the same number of layers. Adversarial samples were generated using `feature-attack` with $k = 3$. Transfer of adversarial samples across transformers progressively becomes poorer as number of layers increases.

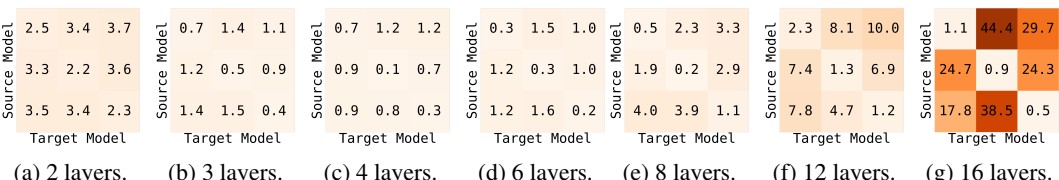

(a) 2 layers.    (b) 3 layers.    (c) 4 layers.    (d) 6 layers.    (e) 8 layers.    (f) 12 layers.    (g) 16 layers.

Figure 20: Same as above figure (19) but adversarial samples were generated using `label-attack` with $k = 7$. As with `feature-attack`, transfer of adversarial samples samples across transformers progressively becomes poorer as number of layers increases.

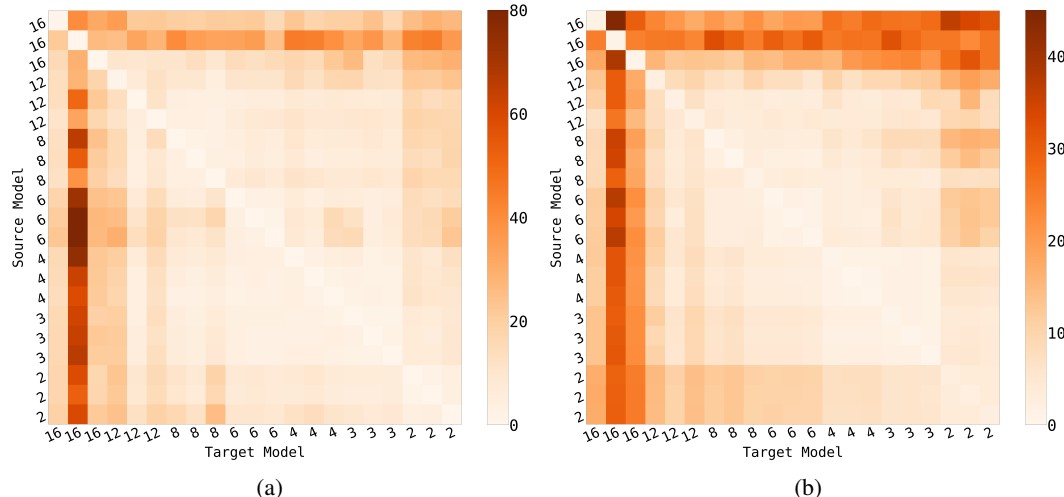

(a)                    (b)

Figure 21: Target Attack Error for different target models on adversarial samples possibly generated using a source model with a different number of layers. In (a) adversarial samples were generated using `feature-attack` with $k = 3$. In (b) adversarial samples were generated using `label-attack` with $k = 7$. Transfer is generally worse when

### B.5    HIJACKING ATTACKS ON ORDINARY LEAST SQUARE

Linear regression can be solved using ordinary least square. This solution can be written in closed-form as follow:

$$\widehat{y} = f(X, Y, x_{\text{query}}) = \left(X^\top X\right)^{-1} X^\top Y x_{\text{query}} \tag{19}$$

where $X = [x_1^\top; x_2^\top; \cdots; x_N^\top]$ and $Y = [y_1, ..., y_N]$. We implement a gradient-based adversarial attack on this solver by using Jax autograd to calculate the gradients $\nabla_X f(X, Y, x_{\text{query}})$ and $\nabla_Y f(X, Y, x_{\text{query}})$. Similar to our gradient-based attack on the transformer, we only update a randomly chosen subset of entries withing $X$ and $Y$. In OLS, $X$ and $Y$ are not tokenized, however, for consistency of language, we will continue to refer to the individual entries of these matrices, i.e., $x_i, y_i$ as tokens. We perform 1000 iterations and use a learning rate of 0.01 for both `feature-attack` and `label-attack`.

Figure 22 shows results for `feature-attack` and y-attack respectively on OLS for $y_{\text{bad}}$ chosen by using $\alpha = 1.0$. The adversarial robustness of OLS is qualitatively similar to that of the transformer; for a fixed compute budget, single-token `label-attack` are much less successful compared to single-token `feature-attack`, and target attack error is lower when greater number of tokens are attacked.

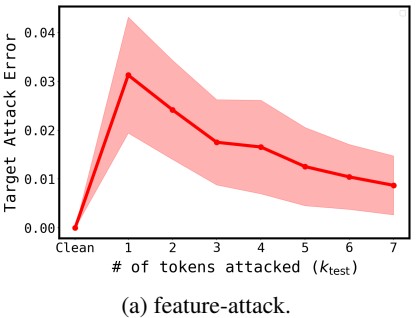 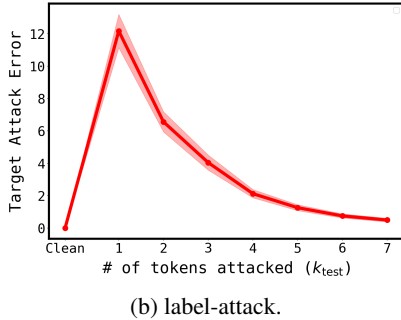

(a) feature-attack.                (b) label-attack.

Figure 22: The adversarial robustness of ordinary least squares to gradient-based hijacking attacks is qualitatively similar to that of the transformers.

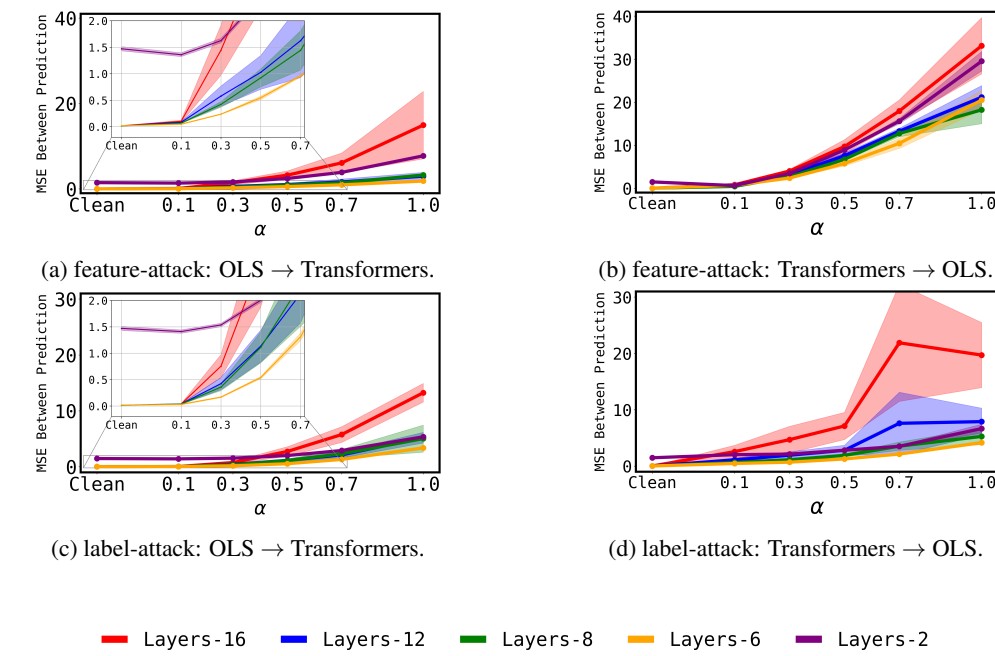

(a) feature-attack: OLS → Transformers.

(b) feature-attack: Transformers → OLS.

(c) label-attack: OLS → Transformers.

(d) label-attack: Transformers → OLS.

Figure 23: The mean squared error between the predictions being made by the transformer and OLS on adversarial samples tends to increase as the 'OOD-ness' of the $y_{bad}$ increases. Furthermore, the difference in prediction is generally higher when the hijacking attacks are derived using the transformer (notice the differences in scale). For `feature-attack`, we attack 3 tokens and for y-attack we attack 7 tokens when creating adversarial samples.

We further look at the transfer of adversarial attacks between transformers and OLS. Specifically, by attacking OLS we create a set of adversarial samples and then measure the mean squared error (MSE) between the predictions of OLS and different transformers on these adversarial samples, and vice versa. Figure 23 shows the transfer for adversarial samples for different values of $\alpha$ for sampling $y_{bad}$. For `feature-attack`, we attack 3 indices and for y-attack, we attack 7 indices. We can make following observations from this figure: (i) the predictions made by OLS and transformers tend to diverge as $\alpha$ increases. This indicates lack of alignment between the predictions made by OLS and transformers OOD. (ii) For `feature-attack`, MSE between predictions is significantly lower when adversarial samples are sourced by attacking OLS relative to when adversarial samples are sourced by attacking the transformers. In other words, adversarial samples transfer better from OLS to transformers but not vice versa. This hints at the fact that adversarial robustness of the transformers is worse than that of OLS. (iii) For y-attack, the aforementioned asymmetry in transfer above does not exist except for transformers with layers 16 and 12. (iv) Finally, we note that transformer with 16 layers clearly always behaves in an anomalous fashion, with transformers with layers 12 and 2 also sometimes behaving anomalously, which is in line with the discussion in previous section on intra-transformer transfer of adversarial samples.

In Figure 24, we present complementary results showing MSE between predictions of OLS and transformers on adversarial samples when different number of tokens are attacked for $\alpha = 1.0$. These results further support the observations made in the previous paragraph.

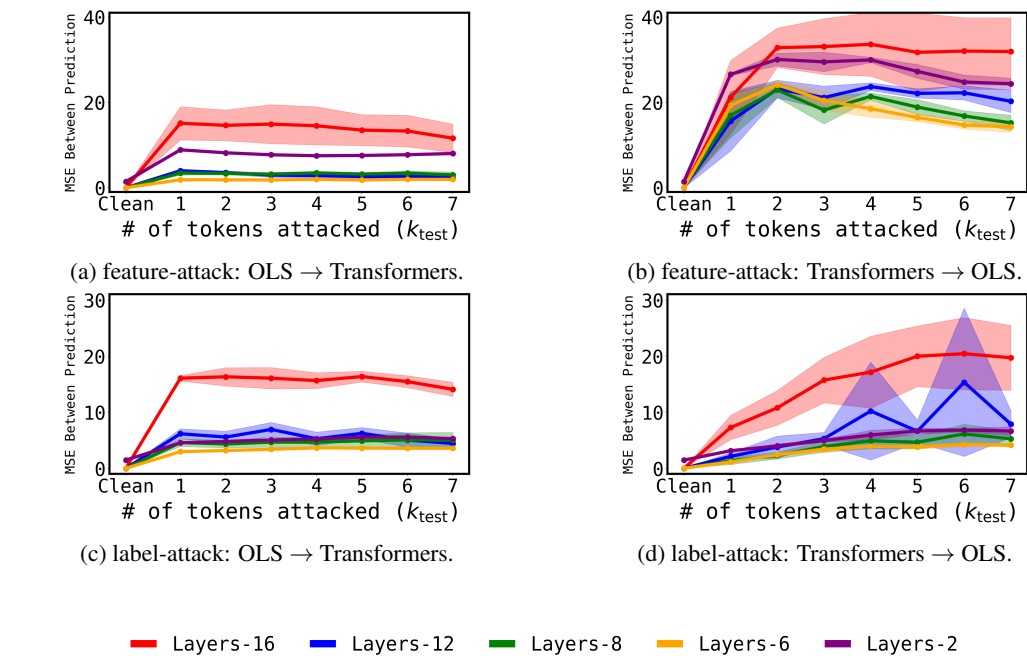

Figure 24: The mean squared error between the predictions being made by the transformer and OLS on adversarial samples tends to be higher when the adversarial samples are sourced by attacking transformers. In the above plot, we use $\alpha = 1.0$ for sampling $y_{\text{bad}}$.

## C  TRAINING DETAILS AND HYPERPARAMETERS

### C.1  LINEAR TRANSFORMER

To match the setup considered in Theorem 4.1, we implement linear transformer as a single-layer attention-only linear transformer as described in equation 10. We train the linear transformer for $2M$ steps with a batchsize of $1024$ and learning rate of $10^{-6}$.

### C.2  STANDARD TRANSFORMER

Our training setup closely mirrors that of Garg et al. (2022). Similar to their setup, we use a curriculum where Details of our architecture are given in Table 1. We gave the number of parameters present in various transformer models with different number of layers in Table 2. Important training hyperparameters are mentioned in Table 3.

| Parameter | Value |
|---|---|
| Embedding Size | 256 |
| Number of heads | 8 |
| Positional Embedding | Learned |
| Number of Layers | 8 (unless mentioned otherwise) |
| Causal Masking | Yes |

Table 1: Architecture for the transformer model.

| Number of Layers | Parameter Count |
|---|---|
| 2 | $1,673,601$ |
| 3 | $2,463,553$ |
| 4 | $3,253,505$ |
| 6 | $4,833,409$ |
| 8 | $6,413,313$ |
| 12 | $9,573,121$ |
| 16 | $12,732,929$ |

Table 2: Hyperparameters used for training transformer models with GPT-2 architecture.

| Hyperparameter | Value |
|---|---|
| Learning Rate | $5 \times 10^{-4}$ |
| Warmup Steps | 20,000 |
| Total Training Steps | 500,000 |
| Batch Size | 64 |
| Optimizer | Adam |

Table 3: Hyperparameters used for training transformer models with GPT-2 architecture.

### C.3 ADVERSARIAL ATTACK AND ADVERSARIAL TRAINING DETAILS

We implement our adversarial attacks as simple gradient descent on the (selected) inputs with the target attack error as the optimization objective. We briefly experimented with variations of gradient descent, e.g., gradient descent with momentum but found those to perform at par with simple gradient descent.

When performing `feature-attack`, we used a learning rate of 1 and when performing `label-attack`, we used a learning rate of 100. When performing `joint-attack`, we used a learning rate of 1 when perturbing x-tokens and a learning rate of 100 when perturbing y-tokens. We chose the learning rates based on best performance within 100 gradient steps. Using lower values of learning rates resulted in proportionally slower convergence, and hence were avoided.

In all our plots, we show results across three different models and use 1000 samples for each model.

**Differences Between Adversarial Attacks and Adversarial Training**: The two major differences in our adversarial traning setup, compared with adversarial attacks setup are:

- During adversarial attacks (done on trained models at test time), we sample $y_{\text{bad}}$ according to the expression 8, but during adversarial training we sample $y_{\text{bad}}$ by sampling a weight vector $w \sim N(0, I_d)$ independent of the task parameters $w_\tau$ and setting $y_{\text{bad}} = w^\top y_{\text{bad}}$.

- During adversarial attacks, we perform 100 steps of gradient descent, but in adversarial training, we only perform 5 steps of gradient descent.

Both the above changes were done to help improve the efficiency of adversarial training.

