# OpenReview forum: "Adversarial Robustness of In-Context Learning in Transformers for Linear Regression"
_ICLR.cc/2025/Conference — Submitted to ICLR 2025_

### Official Review · Reviewer_kCot · 2024-10-29

**Soundness:** 2
**Presentation:** 3
**Contribution:** 2
**Rating:** 5
**Confidence:** 3

**Summary:**

This paper focuses on studying the adversarial robustness of in-context learning for transformers. The paper first proves that a single-layer transformer can be manipulated to output arbitrary predictions, and also uses gradient-based attacks to demonstrate the vulnerability of GPT-2.

**Strengths:**

1. The paper has a solid theoretical analysis to show the vulnerability of single-layer transformers.

2. Beyond single-layer transformers, the paper also considers GPT-2, and the experimental results appear to support the conclusions well.

3. The paper is easy to follow and well-organized.

**Weaknesses:**

1. The novelty of this paper is limited because the phenomenon that single-layer transformers and GPT-2 are vulnerable to adversarial attacks is not surprising, and the paper does not have new and strong technical contribution compared with existing gradient-based attacks.

2. The task that uses GPT-2 for linear regression is not well-motivated. We usually do not use GPT-2 to solve linear regression.

3. The paper does not propose any new defense method beyond adversarial training, which is a standard baseline.

**Questions:**

No question, and please refer to the weakness part.

---

> ### Author Response · Authors · 2024-11-19
> **Authors' Response**
>
> We thank the reviewer for their detailed feedback. We have revised our manuscript significantly to address you and the other reviewers’ concerns; we have summarized these changes in a [global comment](https://openreview.net/forum?id=cnecLUNs6w&noteId=Au8WJmfyd0) to all reviewers.  Below we address your specific comments:
>
> **Novelty of Paper and Technical Significance of Results:**
> The primary novelty of the work is in studying and understanding *how* in-context learning in transformers is non-robust. Specifically:
> - For linear transformers we show they are vulnerable *because* they implement gradient descent - demonstrating how seemingly desirable algorithmic properties can lead to exploitable weaknesses.
> - In GPT2 style transformers, we have shown (with addition of new results in Sec. 5.6 regarding transferability of adversarial attacks from ordinary least squares (OLS) to transformers and vice versa) that these transformers are non-robust in a way that is distinct from both gradient descent and ordinary least squares. This is a novel result as prior works had strongly suggested that these transformers are implementing gradient descent or OLS.
> - Our discovery that attacks transfer readily between smaller transformers but poorly between larger ones provides the first evidence of non-universal learning mechanisms in architecturally identical transformers.
> - The success of adversarial training is surprising given the historical difficulty of robust regression with adaptive adversaries - a problem that saw meaningful progress only in the last decade (citation)
> - The generalization of adversarial training from K to K' > K examples suggests transformers can learn remarkably robust algorithms through simple training procedures, and also illustrates transformer’s ability to perform non-convex optimization in-context.
>
> Please also see our [global comment](https://openreview.net/forum?id=cnecLUNs6w&noteId=Au8WJmfyd0) where we discuss the relevance of this work to various sub-communities within machine learning.
>
> **Reason for studying regression setting**:
> - Transformers trained to do in-context learning are increasingly being used for tabular data settings including regression ([Requeima et al., 2024](https://arxiv.org/abs/2405.12856), [Hollmann et al. 2022](https://arxiv.org/abs/2207.01848), [Ashman et al. 2024](https://arxiv.org/abs/2406.13493)). This is part of a general trend where in-context learning capabilities are being utilized to develop performant models across various domains, e.g., vision, reinforcement learning and robotics ([Zhang et al. 2023](https://arxiv.org/abs/2301.13670),[Elawady et al. 2024](https://arxiv.org/abs/2410.02751), [Raparthy et al. 2023](https://arxiv.org/abs/2312.03801)).
> - Additionally, studying a more abstract setting like regression enables developing a more robust understanding of a transformer's behavior, e.g., it allowed us to theoretically show and diagnose the vulnerability of linear transformers to hijacking attacks. At the same time, we would like to stress that robust regression is a highly challenging problem with computationally efficient methods being developed for this problem only in the last decade. Thus, transformers' ability to perform robust regression is both surprising as well as likely to be of interest to the robust statistics community.
> - Studying in-context learning capabilities of transformers using linear regression is also not unprecedented and is in line with the number of prior works ([Akyürek et al. 2022](https://arxiv.org/abs/2211.15661), [Garg et al. 2022](https://arxiv.org/abs/2208.01066), [Oswald et al. 2023](https://proceedings.mlr.press/v202/von-oswald23a.html), [Zhang et al. 2024](https://arxiv.org/abs/2306.09927), [Ahn et al. 2023](https://arxiv.org/abs/2306.00297)).
>
> **New defenses**: Given that standard adversarial training already works well in our setting without significant performance degradation, developing novel defense methods does not seem necessary. We would again stress that given the historical difficulty of robust regression in the presence of an adaptive adversary, the success of adversarial training in this setup and especially the efficiency of adversarial training is a surprising finding.
>
> Please let us know if you have any further questions. We would be happy to address any remaining concerns or expand on any of these points.

---

> ### Author Response · Authors · 2024-11-23
> **Request for Response on Authors' Response**
>
> Respected reviewer, we have given a detailed response to your comments below (and in the [global response](https://openreview.net/forum?id=cnecLUNs6w&noteId=Au8WJmfyd0)). As there are only few days remaining in the discussion period, we would greatly appreciate if you could review our response and let us know if you have any further questions. If we have successfully addressed your concerns, we request that you please revise your score accordingly.

---

> > ### Comment · Reviewer_kCot · 2024-11-28
> > **Reply to Rebuttal**
> >
> > I appreciate the authors' effort to address the concerns. However, after reading other reviews, I decide to keep my rating because the motivation of using LLMs for a very simple task such as regression is still unclear. This is not a standard way of using LLMs.

---

### Official Review · Reviewer_g6bB · 2024-11-01

**Soundness:** 3
**Presentation:** 3
**Contribution:** 1
**Rating:** 6
**Confidence:** 3

**Summary:**

The paper shows that linear transformers trained on linear regression aren't robust to hijacking attacks. Moreover, the authors show that these attacks don't transfer from small models to more complex GPT2-style ones (and even among them). The latter can be hijacked with gradient-based optimization. Finally, adversarial training (or fine-tuning) is shown to be promising to prevent these attacks.

**Strengths:**

- Methods and experiments are clearly explained and carried out. The work is original, but the scope could have been better stated.

- Results are well presented and commented on. However, they're not very significant.

- I personally like the part about adversarial training, as it is a promising method to make transformers more robust to attacks.

**Weaknesses:**

- I don't get the point of this work and why it has to be considered relevant for the community. I would say this better, possibly on the first page.

- I would have tried to inspect better why attacks don't work on GPT2 style models. This would have been helpful in understanding better how these models perform linear regression.

**Questions:**

1. What's the meaning of these attacks in the context of linear regression? Is there a more practical interpretation?

2. How do these results compare to when using OLS? Can the latter be used as a baseline?

3. What does adversarial training on attacks mean from a linear regression perspective? (eg. can it be related to some form of regularization?)

---

> ### Author Response · Authors · 2024-11-19
> **Author's Response**
>
> We thank the reviewer for their detailed feedback. We have revised our manuscript significantly to address you and the other reviewers’ concerns; we have summarized these changes in a global comment to all reviewers.  Below we address your specific comments:
>
> **Relevance to the community**: Please see our [global response](https://openreview.net/forum?id=cnecLUNs6w&noteId=Au8WJmfyd0) which answers this question in detail. As suggested by you, we have also revised our introduction and discussion sections to make the contributions of this work, and their implications, more salient.
>
> **Meaning of attacks in linear regression context**
> These attacks can be thought of as robustness to adversarial outliers, either in the feature/data space ($x$), the label space ($y$), or both ($z=(x,y)$).  Equivalently, these can be viewed as robustness to data-poisoning attacks.
>
> **Comparison with OLS**
> Our new results in Section 5.6 show that alignment between OLS and transformers weakens significantly out-of-distribution, with transformers exhibiting additional adversarial vulnerabilities relative to OLS.
>
> **Meaning of adversarial training from linear regression perspective**
> This is a great question.  For GPT2-trained transformers, our experiments strongly suggest that they don’t implement any standard regression algorithm - for instance, ridge regression produces a solution which is linear in the labels, which would thus be susceptible to “y-attacks” (renamed label attacks) in the same way that the linear transformer is, but our Figure 3 demonstrates this is not the case.  Indeed, there are very few known algorithms for linear regression which are robust to hijacking attacks, and such algorithms often require complex routines to mitigate the effects of (unknown) adversarial examples ([Cherapanamjeri et al., 2020](https://arxiv.org/abs/2007.08137)).  We think it would be interesting to develop a mechanistic understanding of what algorithm the adversarially-trained transformers implement.
>
>
> “I would have tried to inspect better why attacks don't work on GPT2 style model…”
> Prior work pointed to the possibility that GPT2 models implemented either gradient descent ([Oswald et al. 2023](https://proceedings.mlr.press/v202/von-oswald23a.html), [Zhang et al. 2024](https://www.jmlr.org/papers/v25/23-1042.html)) or ordinary least squares for regression problems ([Akyürek et al. 2022](https://arxiv.org/abs/2211.15661), [Garg et al. 2022](https://arxiv.org/abs/2208.01066)). Our results demonstrate this is not the case, since attacks derived from gradient descent (and our Theorem 4.1 shows linear transformers implement gradient descent) don’t transfer to GPT2, and attacks from GPT2 don’t transfer to OLS (our new Section 5.6).  This shows which algorithms GPT2 does not implement, but we aren’t confident about which algorithm it does implement, and we think this is an interesting direction for future research.
>
> Please let us know if you have any further questions. We would be happy to address any remaining concerns or expand on any of these points.

---

> ### Author Response · Authors · 2024-11-23
> **Request for Response on Authors' Response**
>
> Respected reviewer, we have given a detailed response to your comments below (and in the [global response](https://openreview.net/forum?id=cnecLUNs6w&noteId=Au8WJmfyd0)). As there are only few days remaining in the discussion period, we would greatly appreciate if you could review our response and let us know if you have any further questions. If we have successfully addressed your concerns, we request that you please revise your score accordingly.

---

> > ### Comment · Reviewer_g6bB · 2024-11-27
> >
> > I appreciate the authors’ response and the effort to address the concerns raised. Although I still find the scope of this work to be fairly limited due to its constrained setting and low transferability to larger models, I am increasing my score to 6 and now (marginally) supporting the acceptance of this paper.

---

> > > ### Author Response · Authors · 2024-11-27
> > > **Thanks**
> > >
> > > Thank you for reviewing our response and responding positively to it. We highly appreciate that.

---

### Official Review · Reviewer_8XUG · 2024-11-06

**Soundness:** 2
**Presentation:** 2
**Contribution:** 2
**Rating:** 3
**Confidence:** 3

**Summary:**

This paper explores the adversarial robustness of in-context learning in transformers, focusing on hijacking attacks in linear regression tasks. Single-layer linear transformers are shown to be easily manipulated, while complex models like GPT-2 resist simple attacks but remain vulnerable to gradient-based ones. Adversarial training enhances model robustness, especially during fine-tuning, and hijacking attacks transfer only among smaller models.

**Strengths:**

1. The paper is overal clear and well-written with sufficient experiments.

**Weaknesses:**

1. I find the use of x-attack, y-attack, z-attack to be confusing. The joint attack (z-attack) does not really corespond to a different dimension to attack. It is better to call it data, label, and joint attack, or something similar.
2. From the motivation level, I am uncertain if hijacking attack is the main concern in the applications of LLM for transformers. These line of work along Garg et al. are used to demonstrate the in-context capability of transformers which is a simplified setting but sufficient. The typical adversarial learning considers the setting where the input is modified only slightly, but the adversary can have large impact on the output. I don't see how this classical analysis can easily transfer to the text domain. How to define the allowable small perturbation. If we naively apply on token-level, people can easily find it. If the text contains number, and we only make small modification on the number, will it actually lead to different output. So, I am not sure if the paper can provide many insights into understanding the hijack attack for LLM. I will raise my score if the author sufficiently addresses this issue.
3. Since the authors mainly investigate the in-context learning setup, they should consider some context-specific attacks, for example within context or across-context. I don't think uniform attack on all possible entries are the only interesting scenario.
4. I appreciate the empirical effort by the authors. In terms of novelty, I would like to see more novel algorithm design and potential implications, since incontext is still a different setting from the standard ML setting.
5. I find the experiment section to be hard to follow. Even though the model choice is discussed in the setup section, it is unclear what architecture and training procedure that each figure corresponds to e.g. figure 2.3.4.

**Questions:**

1. Can the author provide why the adversarial examples fail to transfer? It is possible that the attack examples cannot be transferred from linear attention model to GPT2, but it can transfer from GPT2-small to GPT2-large.
2. Has the author experiments with larger model than GPT2, such as llama 8B? I am wondering if the hijack attacks can be mitigated by the emergent ability of transfoermers.
3. Did the author experiements with TRADES, which usually serves as a better defense than the original PGD.

---

> ### Author Response · Authors · 2024-11-19
> **Authors' Response (Part 1)**
>
> We thank the reviewer for their detailed feedback. We have revised our manuscript significantly to address you and the other reviewers’ concerns; we have summarized these changes in the [global comment](https://openreview.net/forum?id=cnecLUNs6w&noteId=Au8WJmfyd0) to all reviewers. Below we address your specific comments:
>
> **Motivation for and relevance of hijacking attacks**:
> In LLMs, hijacking attacks can be considered to be an instance of prompt-injection / prompt-manipulation attacks which are now widely recognized as serious security vulnerabilities (e.g., [Anwar et al. 2024](https://arxiv.org/abs/2404.09932), Section 3.5). For sequence-based models like transformers/LLMs, prompt-manipulation is a natural attack vector through which an adversary can influence a model’s prediction by adding adversarial tokens into the context. In this setting, a small change of the input corresponds to a small number of tokens added/changed in a long prompt. Transformers are increasingly being used in critical settings which rely upon in-context learning to perform their task, e.g., clinical decisions where the patient history and data is given as context ([Nori et al. 2023](https://arxiv.org/abs/2311.16452)), robotic control algorithms which rely on ICL ([Elawady et al. 2024](https://arxiv.org/abs/2410.02751)), automated code completion ([Patel et al. 2024](https://arxiv.org/abs/2311.09635)), etc., and in many settings the end-user/adversary doesn’t have the ability to manipulate the internals of the transformer but does have the ability to append tokens to the input of e.g,. an API. As such, we believe understanding the robustness of in-context learning under hijacking attacks is a particularly relevant problem, since they could change the outcome of a clinical decision, manipulate the movement of a robot’s arms, insert code backdoors, and so on.
>
> We would also like to emphasize that hijacking attacks on LLMs have already been demonstrated in prior work, e.g.,
> - [Qiang et al, 2023](https://arxiv.org/abs/2311.09948) give a practical attack on LLMs that closely mirrors our x-attack (now renamed featuredata attack) in construction and show that their attack results in LLM outputting the target token.
> - [Bailey et al. 2023](https://arxiv.org/abs/2309.00236) give a hijacking attack on vision-language models (VLM) where they manipulate images present in the context to control the behavior of VLM.
>
> Note that applications of in-context learning in transformers (and hence dangers of hijacking attacks) are not just restricted to LLMs. They are seeing rapid adoption in many domains, e.g., vision ([Zhang et al. 2023](https://arxiv.org/abs/2301.13670), [Kirsch et al. 2022](https://arxiv.org/abs/2212.04458)) and robotics/reinforcement learning ([Elawady et al. 2024](https://arxiv.org/abs/2410.02751), [Raparthy et al. 2023](https://arxiv.org/abs/2312.03801)) with both continuous and discrete input representations.
>
> Our [global comment](https://openreview.net/forum?id=cnecLUNs6w&noteId=Au8WJmfyd0) further goes into the details of the novelty and relevance of our contributions.
>
> **Why adversarial examples fail to transfer**:
> As we mention in our global comment, we provide new experiments in Section 5.6 which further probe transferability of attacks between transformers and OLS. Together with prior results shown in Figure 1 and Figure 6, we have a fairly complex picture of transferability:
> - Within GPT2-style transformers, attacks transfer well across small and medium transformers but transfer poorly from and to larger transformers – even between standard transformers with identical architectures.
> - Attacks derived from the linear transformer (i.e., GD solution) do not transfer well to any GPT-2 style transformers
> - attacks derived from OLS transfer to medium-scale transformers relatively well but not larger transformers (Fig. 7)
> - attacks derived from all transformers transfer poorly to OLS.
> This suggests that GPT2's out-of-distribution behavior differs fundamentally from both gradient descent and OLS, with some shared vulnerabilities to attacks from OLS but vulnerable to a broader set of attacks than OLS is. This challenges prior theories about what algorithms these models implement.

---

> ### Author Response · Authors · 2024-11-19
> **Authors' Response (Part 2)**
>
> **Testing on larger models like LLaMA**
> While computational constraints prevent us from training 8B+ parameter models, our analysis of scaling effects (Section 5.3 and Appendix B.1) suggests that model scale alone does not improve robustness to these attacks.
>
> **TRADES vs standard adversarial training**
> Given that standard adversarial training already works well without significantly impacting clean performance (Figure 5), we found more sophisticated approaches unnecessary for this setting.
>
> **Clarity on x/y/z terminology and experiment details**: We are sorry for the lack of clarity regarding terminology and the experiment section.  We have updated our paper to change the x/y/z attack labels to feature, label, and joint attack.  We have also updated our figures to be more explicit about which architecture and training procedure we use.
>
> **Context-specific attacks**: We are unsure what the reviewer means here and would appreciate further clarification.  In our setting, every attack is context-specific, since the adversary looks at the sequence of tokens and based on that, then adds or perturbs one or more tokens.
>
> We would be happy to clarify any remaining concerns. We hope these revisions and explanations address your questions about the motivation and clarity of our work.

---

> ### Author Response · Authors · 2024-11-23
> **Request for Response on Authors' Response**
>
> Respected reviewer, we have given a detailed response to your comments below (and in the [global response](https://openreview.net/forum?id=cnecLUNs6w&noteId=Au8WJmfyd0)). As there are only few days remaining in the discussion period, we would greatly appreciate if you could review our response and let us know if you have any further questions. If we have successfully addressed your concerns, we request that you please revise your score accordingly.

---

> > ### Comment · Reviewer_8XUG · 2024-11-27
> > **Still unclear relationship to standard LLM**
> >
> > I thank the author for the detailed response.
> >
> > However, I still don't find the paper to be well-motivated. Empirical paper like Qiang et al, 2023, Bailey et al. 2023, indeed give a similar feature-attack, but the attack is operating on the text domain with potentially more complex sequence-dependency. The author's experiment is on the classical in-context finetuning setting where the data is structured in a constrained way [x1, y1; x2, y2 ; ...], which does not consider realistic complication like task vector, or input with semantic meaning. I don't see if the results on this neat setting can transfer in any way to the standard LLM setting.
> >
> > I acknowledge that proving the existence of hijack attack on the classical Garg et al. / Zhang et al. setting is interesting, but the connection to standard LLM is still unclear. The authors should consider actually running Qiang 2023 or some generalized version of it to feature/label/joint attack, and show that if the phenomenon on larger setting and on ICL finetuning setting are same or different. Then, the author can investigate about why it happens.
> >
> > Thus, I will keep my current score.

---

> > > ### Comment · Reviewer_8XUG · 2024-11-27
> > > **Attack a full example/demonstration is unrealistic**
> > >
> > > Furthermore, Qiang et al. 2023 attack by adding an adversarial suffix, which is much weaker than an arbitrary attack on X. Imagine in real in-context learning scenario, the demonstration feature data is a long paragraph that describes certain math problem. The authors attack will pretty much replace this problem with some arbitrary adversarial text, then any user can simply pick it out.
> > >
> > > By "context-specific", I mean attack multiple examples at the same time, specifically, attacking multiple demonstrations in the same context. This way, you don't need to give arbitrary example that can simply picked out.

---

> ### Comment · Reviewer_8XUG · 2024-11-27
>
> I have taken account of the global comment, and I keep my current rating. If you are very into argument, I suggest you submitting to a more theoretical-driven conference.

---

### Author Response · Authors · 2024-11-19
**Authors' Global Response -- New Results and Clarification on Contributions**

We are thankful for all of the reviewers’ constructive and useful feedback, which has helped us to strengthen our work. Based on your comments, we have made several substantial improvements to clarify our key contributions and their broader significance. We have colored the major changes in blue for ease of reviewers. Two key updates are:
- New experimental results comparing transformer behavior to ordinary least squares (Section 5.6 and Appendix B.5).
- Revamped introduction to better motivate our paper, and expanded discussion of implications for different research communities.

## New Experimental Results
We have added new results investigating the transfer of adversarial examples from ordinary least squares (OLS) to transformers and vice versa. These new results show that
- Attacks derived from OLS transfer relatively well (but not perfectly) to medium-scale transformers but do not transfer at all to larger transformers.
- Attacks derived from all transformers transfer poorly to OLS.
These new results imply that transformer’s out of distribution behavior does not match with OLS, challenging prior results which suggest transformers may be implementing OLS.

## Relevance and Motivation of This Work
As transformers and large language models increasingly serve as the foundation for real-world applications, understanding how they perform in-context learning becomes crucial for both reliability and security. Our study of hijacking attacks - where an adversary can manipulate model behavior through carefully crafted context examples - reveals fundamental vulnerabilities in this learning mechanism. These findings are particularly relevant as ICL-based inference becomes prevalent across domains including embodied AI ([DeepMind ‘23](https://arxiv.org/abs/2301.07608), [Elawady et al. 2024](https://arxiv.org/abs/2410.02751)), and structured data (tabular data) processing ([Requeima et al., 2024](https://arxiv.org/abs/2405.12856), [Hollmann et al. 2022](https://arxiv.org/abs/2207.01848), [Ashman et al. 2024](https://arxiv.org/abs/2406.13493)), and is used for sensitive applications like clinical decision making ([Nori et al. 2023](https://arxiv.org/abs/2311.16452)) and robot control [Elawady et al. 2024](https://arxiv.org/abs/2410.02751).

We believe our work reveals several fundamental insights about transformer behavior that is likely to be of interest to multiple research communities:

**For researchers studying in-context learning:** Our use of hijacking attacks to examine alignment between transformers and traditional solvers is novel and resulted in novel findings.
- Our transfer analysis reveals that transformers' out-of-distribution behavior is mechanistically distinct from both gradient descent (which is implemented by linear transformers) and OLS. This  challenges prior explanations of how these models perform in-context learning ([Akyürek et al. 2022](https://arxiv.org/abs/2211.15661), [Garg et al. 2022](https://arxiv.org/abs/2208.01066), [Oswald et al. 2023](https://proceedings.mlr.press/v202/von-oswald23a.html), [Zhang et al. 2024](https://arxiv.org/abs/2306.09927), [Ahn et al. 2023](https://arxiv.org/abs/2306.00297)).
- Our discovery that adversarial attacks transfer readily between smaller transformers but poorly between larger ones provides the first evidence of non-universal learning mechanisms in architecturally identical transformers.

Combined, these results indicate that developing an understanding of in-context learning even in a highly structured setting like linear regression is an open problem and may be more challenging than previously thought to be.

**For the robust statistics community:**
Achieving adversarial robustness against adaptive adversaries in regression is historically very difficult, with meaningful progress only being made in the last decade ([Diakonikolas and Kane, 2019](https://arxiv.org/abs/1911.05911)) or so. Our finding that transformers can learn robust algorithms through simple adversarial training that generalizes beyond the training setting is thus particularly noteworthy and likely of interest to the robust statistics community.

**For security and adversarial robustness community**:
Hijacking attacks on LLMs through perturbation of in-context dataset have already been demonstrated in prior work, but not well-understood ([Qiang et al. 2023](https://arxiv.org/abs/2311.09948), [Bailey et al. 2023](https://arxiv.org/abs/2309.00236)). On an abstract task, we have studied how these vulnerabilities arise resulting in two notable findings:
- These vulnerabilities arise due to the underlying algorithmic mechanisms of ICL
- Even adversarially trained networks are vulnerable to deeper attacks (that target more tokens).

---

### Author Response · Authors · 2024-11-25
**Discussion Period Ending Soon**

Dear reviewers, there is only one day left in the discussion period. If you have any unaddressed concerns, we would be thankful if you could share them and give us a chance to respond. If not, we hope that you may consider updating your scores.

---

### Meta-Review · Area_Chair_S8aV · 2024-12-21

**Metareview:**

The submission "Adversarial Robustness of In-Context Learning in Transformers for Linear Regression" investigates the question of robustness of in-context-learning in transformer models against small-scale modifications of the input context. The main vehicle of their analysis is a proof that that 1-layer transformers, which approximate gradient descent with ICL (as based on previous work in connecting ICL and optimization strategies), are non-robust because of this.
The authors then run a series of transfer experiments, between linear and normal tranformer models, OLS models, models of different seeds and sizes, and models trained with adversarial training.

Ultimately, the submission is unable to convince the reviewers and me that this is meaningful. The proof is an interesting niche, but hard to draw conclusions to more practical systems, where other formulations, such as Wolf et al, "Fundamental Limitations of Alignment in Large Language Models" already prove stronger theoretical statements about the existence of prompt injection attacks. There are some applications of the results as analysis tools to understand ICL algorithms, but the submission does not find a substantial conclusion.

Further, the submission notes that these findings call into question previous results about the algorithms approximated by ICL, but previous work makes no statement whether the instantiation of the algorithm is adversarially robust.


I think the authors could improve this work by consolidating their findings into a more consistent presentation. Based on this, I do not recommend acceptance.

**Additional Comments On Reviewer Discussion:**

During the discussion, the reviewers raise the unclear applicability of the presented results as main concern. The authors pressure the reviewers to raise their scores anyway, but provide no new substantial arguments in favor of their submission.

---

### Decision · Program_Chairs · 2025-01-22

Reject